

# Seasonality of the Quasi-biennial Oscillation signal in water vapor in the tropical stratosphere

Qian Lu[1, 3], Jian Rao[1, *], Chunhua Shi[1], and Chaim I. Garfinkel[2]

[1]Collaborative Innovation Center on Forecast and Evaluation of Meteorological Disasters / Key Laboratory of Meteorological Disaster of Ministry of Education, Nanjing University of Information Science and Technology, Nanjing 210044, China
[2]Fredy and Nadine Herrmann Institute of Earth Sciences, The Hebrew University of Jerusalem, Givat Ram, Jerusalem, Israel
[3]China Meteorological Administration Xiong'an Atmospheric Boundary Layer Laboratory, Xiong'an New Area 071800, China

*Correspondence to*: Dr. Jian Rao (raojian@nuist.edu.cn)

**Abstract.** Stratospheric water vapor is an important greenhouse gas, which affects the radiation balance and temperature structure of the stratosphere and troposphere. Although previous studies have investigated the water vapor variability associated with the quasi-biennial oscillation (QBO), the seasonal difference in the water vapor QBO are still not well understood. Using the ERA5 reanalysis and SWOOSH observations, this study compares the stratospheric water vapor distribution in northern winter and summer under different QBO phases. The water vapor and zonal winds are positively correlated in the mid-to-lower stratosphere; however this relationship weakens in the northern summer. The mean vertical transport term via the QBO related residual circulation is the leading factor controlling the water vapor distribution in most of the stratosphere. This dynamic transport of water vapor in the lower stratosphere by the residual circulation is larger in boreal winter than in summer. Further, the dehydration effect by cold temperature in the lower stratosphere is also more effective in boreal winter than in summer. Tropical deep convection exhibits opposite variations for a given QBO phase in boreal winter versus summer especially over the Indo-Pacific Oceans. This further enhances the temperature difference between the QBO easterly and westerly phases in winter and reduces the temperature contrast in summer. It is still a challenge for models to reproduce the water vapor QBO: CMIP6 models tend to underestimate the water vapor QBO amplitude, and the seasonal contrast in the water vapor QBO between boreal winter and summer is underrepresented in most models.

## 1 Introduction

Water vapor is the dominant greenhouse gas in the atmosphere (Dessler et al., 2013; Solomon et al., 2010), and stratospheric water vapor more specifically affects the radiative balance and temperature structure of the stratosphere (Bi et al. 2011; Xia et al. 2021). It can also further affect the stratospheric circulation through the mutual adjustment relationship between thermodynamic balance and dynamic balance (Banerjee et al., 2019; Charlesworth et al., 2023; de Forster and Shine, 1999).



In addition, it also modulates chemical processes in the stratosphere, such as ozone depletion (Tian et al., 2009, 2023; Wohltmann et al., 2024).

Water vapor can enter the stratosphere through different channels (Lu et al., 2020; Mote et al., 1996; Randel et al., 2015; Randel and Park, 2019; Tinney and Homeyer, 2023; Yue et al., 2019). The primary channel through which tropospheric water vapor enters the stratosphere is the tropical path (Evan et al., 2015; Garfinkel et al., 2013; Mote et al., 2000; Randel and Park, 2019), where cold temperatures at the tropopause determine the tropospheric water vapor content entering the stratosphere (Hardiman et al., 2015; Xia et al., 2019). These cold temperatures freeze and condense most of the water vapor that approaches the tropopause. As a consequence, most of the water that enters the tropopause transition layer falls back to the troposphere, and only a tiny part of the water vapor eventually reaches the stratosphere (Brewer, 1949; Dessler et al., 2013; Holton and Gettelman, 2001; Ueyama et al., 2016; Xia et al., 2019).

As a leading mode of interannual variability in the equatorial stratosphere, the quasi-biennial oscillation (QBO) completes a cycle every 28 months on average (Baldwin et al., 2001; Rao et al., 2020a). During a QBO cycle, alternating easterly and westerly winds propagate downwards from the equatorial upper stratosphere to the tropopause (Baldwin et al., 2001; Cai et al., 2022; Rao et al., 2020b, 2023a, b). The QBO is primarily driven by wave fluctuations of different scales in the tropics. Waves propagating upward carry zonal momentum, and the momentum of westerly or easterly winds is deposited into the stratosphere after wave breaking (Coy et al., 2017; Lindzen and Holton, 1968; Wang et al., 2023). It is observed that the intensity of the QBO easterly phase (30–35 m s$^{-1}$) is usually stronger than that of westerly phase (15–20 m s$^{-1}$), with the QBO amplitude maximized around 20–30 hPa (Anstey et al., 2022; Baldwin et al., 2001). The maintenance and transition of the QBO westerly and easterly phases are related to tropical wave activities, including Kelvin waves, mixed Rossby–gravity waves, and internal gravity waves (Canziani and Holton, 1998; Randel and Wu, 2005; Richter et al., 2014; Holt et al., 2016; Kang et al., 2020; Bramberger et al., 2022; Garfinkel et al., 2022; Pahlavan et al., 2023).

The evolution of the tropical stratospheric QBO is accompanied by the downward propagation of temperature anomalies (Baldwin et al., 2001; Rao et al., 2020a), which directly affects the freezing temperature of water vapor entering the stratosphere (Hardiman et al., 2015; Tao et al., 2015). The QBO winds are in balance with an anomalous secondary circulation, which both modulates the BD circulation and affects the distribution of chemical components (e.g., ozone, methane, water vapor) in the stratosphere (Baldwin et al. 2001). This anomalous secondary circulation can also explain the stratospheric temperature anomalies associated with adiabatic motions (Baldwin et al., 2001; Lu et al., 2020; Rao et al., 2019), which in turn affect water vapor entering the stratosphere (Tian et al., 2023; Xia et al., 2021; Ziskin Ziv et al., 2022). Previous work has evaluated the QBO–water vapor effect in a subset of CMIP6 models, and found they qualitatively capture but underestimate the QBO effect (Ziskin et al. 2022). The QBO leads to a change of static stability in the tropical lower stratosphere and tropopause, further adjusts deep convection activities, and affects the upward transport of tropical water vapor (Dong et al., 2020; Tselioudis et al., 2010).

The relationship between the QBO and tropical stratospheric water vapor has been widely investigated (Chen et al., 2005; Tao et al., 2015; Xia et al., 2021). However, it still remains unclear whether the effects of the QBO on stratospheric water



vapor differ between northern winter and summer. The seasonality of the water vapor QBO signal has been seldom studied. This study uses more samples based on the long time series of the QBO signal in stratospheric water vapor and discusses the differences in stratospheric water vapor distribution between different QBO phases and between different seasons. Possible causes of those differences are diagnosed, and the performance of climate models in capturing the QBO signal in water vapor is also evaluated (Ye et al., 2018; Ziskin Ziv et al., 2022).

The organization of this article is as follows. In section 2, a brief description of datasets and methods is given. Section 3 presents the timeseries of the water vapor QBO. Section 4 compares differences in the distribution of stratospheric water vapor anomalies associated with the QBO between boreal winter and summer. Possible mechanisms responsible for the seasonal difference in the stratospheric water vapor signal are also discussed. Section 5 evaluates the simulation of the stratospheric water vapor QBO by CMIP6 models. Finally, conclusions are provided in section 6.

**2 Data and methods**

*a. Datasets*

To investigate the stratospheric water vapor QBO in the tropical stratosphere, the European Centre for Medium-Range Weather Forecasts' fifth generation reanalysis (ERA5) from 1960 to 2020 (Hersbach et al., 2020) was used. The horizontal resolution of the ERA5 reanalysis is 0.25° (latitude) × 0.25° (longitude), and the model has 137 levels in the vertical

direction from 1000 to 0.01 hPa. A horizontal resolution of 1° (latitude) × 1° (longitude) at 37 pressure levels from 1000 to 1 hPa in the vertical direction was collected for this study. The variables used include zonal and meridional wind, specific humidity, and air temperature on pressure levels.

The Stratospheric Water and Ozone Satellite Homogenized (SWOOSH) dataset version 2.7 is also used in this paper. It is a merged record of stratospheric ozone and water vapor measurements taken by a number of limb sounding and solar

occultation satellites over the previous ~40 years (1984 to present). This dataset is a combination of different data sources, including SAGE-II/III/ISS, UARS HALOE, UARS MLS, Aura MLS, ACE-FTS, and OMPS-LP (Davis et al., 2016). Since the UARS HALOE observation data began on 19 October 1991, we have used SWOOSH data from 1992 onwards.

To understand the possible climatic effects of stratospheric water vapor and its representation in models, this study also evaluates the historical simulations of water vapor QBO from 18 models with internally generating QBO from Phase 6 of the

Coupled Model Intercomparison Project (CMIP6) (Eyring et al., 2016; Simpkins, 2017). They are ACCESS-CM2, AWI-CM-1-1-MR, BCC-CSM2-MR, CESM2-WACCM-FV2, CESM2-WACCM, CNRM-CM6-1, E3SM-1-0, E3SM-1-1, EC-Earth3-VEG, EC-Earth3, GFDL-ESM4, HadGEM3-GC31-LL, HadGEM3-GC31-MM, IPSL-CM6A-LR, MIROC6, MPI-ESM1-2-HR, MRI-ESM2.0, and UKESM1-0-LL. The historical simulation is a mandatory historical climate experiment from 1850 through 2014 under all observation-based and time-varying forcings (i.e., greenhouse gas concentrations, aerosols,

ozone depletion, solar cycles, and land use) (Eyring et al. 2016). We only used the first historical run of the CMIP6 models, and all models provide specific humidity.



*b. Methodology*

The climatology of a variable is calculated as long-term monthly average over the time period from 1960 to 2020. The anomaly refers to the deviation of the monthly data from the monthly climatology with the trend removed for each calendar

100   month. A Butterworth first-order bandpass filter was used to extract the water vapor variations at the period of 15-60 months, mimicking the water vapor QBO (Krishnamurti et al., 1990; Murakami, 1979).

The QBO index is defined as the stratospheric zonal mean zonal wind anomalies over the equator at 5°S - 5°N (Baldwin et al. 2001; Rao et al. 2020a). Considering that the QBO wind variability is maximized around 30 hPa and for many CMIP models the QBO signal is not present below 30 hPa, the QBO index at 30 hPa is employed. QBO events are selected when the QBO

index is greater than 5 (less than -5) m/s, following previous studies (e.g., Rao et al. 2020a, 2020b).

To accurately diagnose the driving factors of the zonal mean distribution of stratospheric water vapor under different QBO conditions, we use the transformed Eulerian-Mean (TEM) tracer continuity equation under spherical z coordinates as follows (Garcia and Solomon, 1983; Monier and Weare, 2011):

$$\frac{\partial \bar{\chi}}{\partial t} = -\frac{\bar{v}^*}{a}\frac{\partial \bar{\chi}}{\partial \phi} - \bar{w}^*\frac{\partial \bar{\chi}}{\partial z} - \frac{1}{\rho_0}\nabla \cdot \boldsymbol{M} + \bar{S}, \tag{1}$$

where $\chi$ is the mixing ratio of water vapor, $\bar{v}^*$ and $\bar{w}^*$ are the horizontal and vertical velocities of the transformed Eulerian mean residual circulation. The residual velocities are calculated as follows (Butchart, 2014; Hardiman et al., 2014):

$$\bar{v}^* = \bar{v} - \frac{1}{\rho_0}\frac{\partial}{\partial z}\left(\rho_0 \frac{\overline{v'\theta'}}{\bar{\theta}_z}\right), \tag{2}$$

$$\bar{w}^* = \bar{w} + \frac{1}{a\cos\phi}\frac{\partial}{\partial \phi}\left(\cos\phi \frac{\overline{v'\theta'}}{\bar{\theta}_z}\right). \tag{3}$$

In Eqs. 2 and 3, $\bar{v}^*$ represents the residual meridional wind component, $\bar{w}^*$ is the residual vertical wind component, $\theta$

denotes potential temperature, $a$ stands for Earth's radius, $\phi$ signifies latitude, and $\rho_0$ indicates air density. The overbar denotes zonal averaging, and prime denotes zonal deviations. $\nabla \cdot \boldsymbol{M}$ is the divergence of the eddy flux vector and represents the eddy transport of water vapor. The components of the eddy flux vector M are defined as follows (Garcia and Solomon, 1983):

$$\boldsymbol{M}^{(\phi)} = \rho_0 \left(\overline{v'\chi'} - \frac{\overline{v'\theta'}}{\bar{\theta}_z}\frac{\partial \bar{\chi}}{\partial z}\right), \tag{4}$$

$$\boldsymbol{M}^{(z)} = \rho_0 \left(\overline{w'\chi'} + \frac{1}{a}\frac{\overline{v'\theta'}}{\bar{\theta}_z}\frac{\partial \bar{\chi}}{\partial \phi}\right). \tag{5}$$

The eddy flux vector represents the mass flux of water vapor by eddies. Finally, $\bar{S}$ in Eq. 1 is calculated as the residual of the other terms, which includes the terms of chemical net production of water vapor, evaporation (or condensation), and water vapor eddy transport generated by small-scale disturbances.





## 3 Stratospheric water vapor QBO behavior

To observe the QBO characteristics of water vapor in the tropical stratosphere, the water vapor anomalies in the tropics were filtered for the periodicities of 15–60 months using a Butterworth first-order bandpass filter (Murakami 1979; Krishnamurti et al. 1990). Figure 1 shows the evolution of water vapor anomalies in the tropical stratosphere over the past 60 years for the ERA5 reanalysis and 30 years for the SWOOSH data. In the ERA5 reanalysis, water vapor in the lower tropical stratosphere around 50–100 hPa presents obvious QBO variations, and the maximum amplitude after bandpass filtering can reach ±0.35

130    ppm. The maximum water vapor anomalies in the lower stratosphere subsequently propagate upward to the middle stratosphere, and reach around 10–30 hPa after a year. The amplitude of the water vapor QBO also gradually weakens during the upward propagation, and the maximum amplitude of the water vapor anomalies at 10–30 hPa are only ±0.15 ppm. The water vapor in the upper stratosphere around 1–5 hPa also exhibits obvious QBO variability (Fig. 1a, 1b). The water vapor QBO from SWOOSH is consistent with ERA5 reanalysis, but the amplitude is stronger (Fig. 1c). Since HALOE started from

1992, the water vapor QBO amplitude in the upper stratosphere between 1–5 hPa has increased, which is also shown in ERA5 reanalysis. Alternately, the lack of a water vapor QBO in the upper stratosphere before 1992 in ERA5 could be a reanalysis artifact.

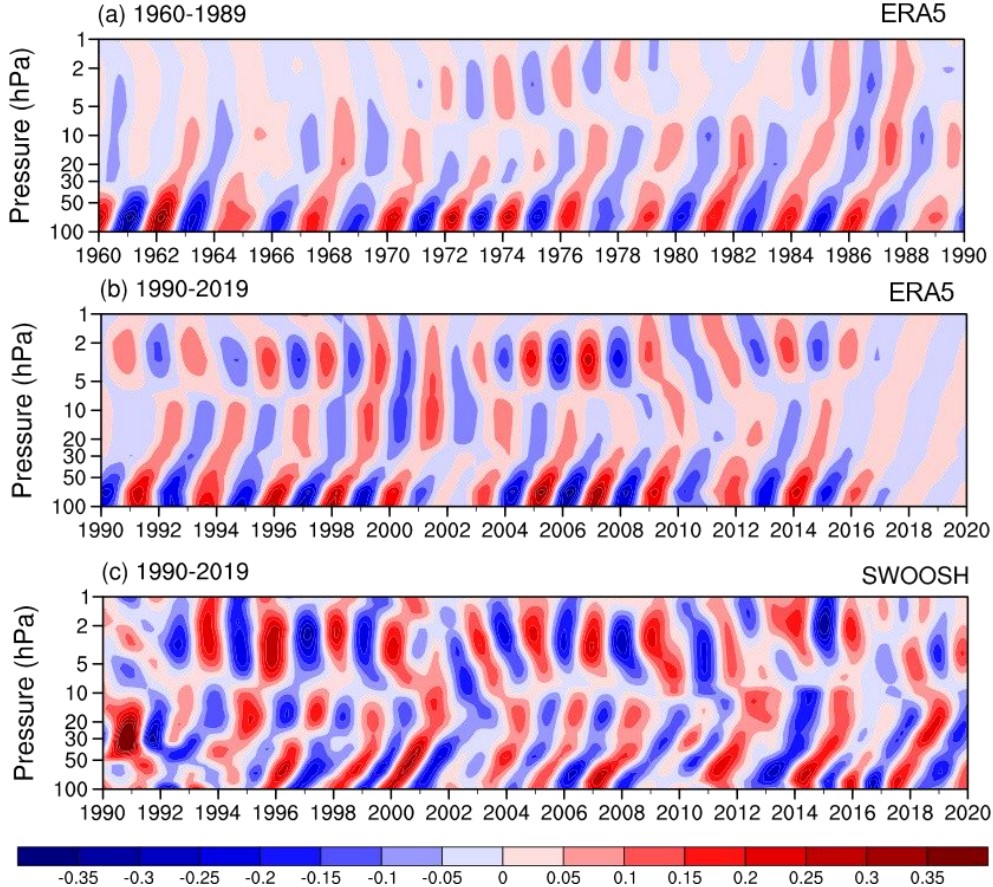



**Figure 1. (a, b) Temporal variations of water vapor anomalies (units: ppm) averaged over the equator (5°S–5°N) with removed linear trends in the tropical stratosphere from 1960–2019 for ERA5 reanalysis. (c) Water vapor anomalies (units: ppm) from 1990–2019 for SWOOSH data. The anomalies are filtered by applying a 15–60-month Butterworth bandpass filter.**

Figure 2 shows the zonal mean zonal wind and temperature anomalies in the tropical stratosphere. The zonal mean zonal wind anomalies in the tropical stratosphere show obvious QBO variations and propagate downward from 5 hPa to the lowermost stratosphere. The amplitude of zonal winds is relatively stable above 15 m/s at 5–30 hPa, and the maximum

central amplitude can exceed 20 m/s. Zonal mean zonal wind anomalies weaken rapidly from 50–100 hPa. Zonal winds alternate between the easterly and westerly most of the time, except for the disruption of the QBO westerly by the lower stratospheric easterly in 2016 (Coy et al., 2017; Wang et al., 2023).

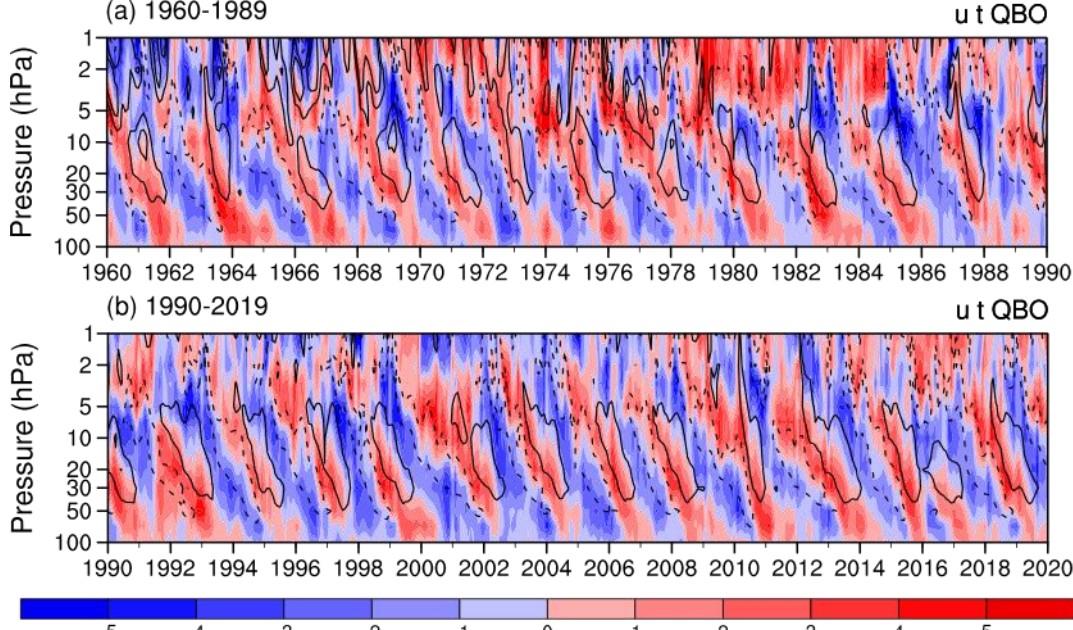

**Figure 2. Temporal variations of zonal mean zonal wind anomalies (contours; units: m/s) and temperature anomalies (shadings;**
**units: K) averaged over the equator (5°S–5°N) with removed linear trends in the tropical stratosphere from 1960–2019.**

Consistent with the zonal wind anomalies, temperature anomalies in the tropics also exhibit evident QBO variability, which gradually propagates downward to the lower stratosphere from 5 hPa. Thermal wind balance predicts that cold anomalies should lie underneath strong easterly winds and warm anomalies underneath westerly winds, and such an effect is clearly evident with peak temperature anomalies in regions of strongest shear as expected theoretically. The temperature anomalies
can propagate downward to 100 hPa, and the temperature anomaly amplitude can exceed 3 K.

The zonal wind and water vapor anomalies at 70 hPa, 10 hPa and 2 hPa are calculated respectively to explore the corresponding relationship between the zonal mean zonal wind QBO and water vapor QBO. The zonal wind anomalies are positively correlated with the water vapor anomalies at 70 hPa (correlation coefficient, CC=0.51). Basically, an increase in water vapor in the lower stratosphere happens during the QBO westerly, and a decrease in water vapor in the lower





stratosphere appears during the QBO easterly phase (Fig. 3a). Their correlation at 10 hPa is even larger than at 70 hPa, which

suggests that the water vapor QBO is more robustly present in the middle stratosphere (CC=0.58, Fig. 3b). The zonal wind

variation at 2 hPa shows weak QBO signal: the zonal wind variability is relatively chaotic (Fig. 3c). Previous studies

indicated that the cold point temperature determines the tropical water vapor in the stratosphere (Randel and Park, 2019), and

so this relationship between the QBO and water vapor is to be expected.

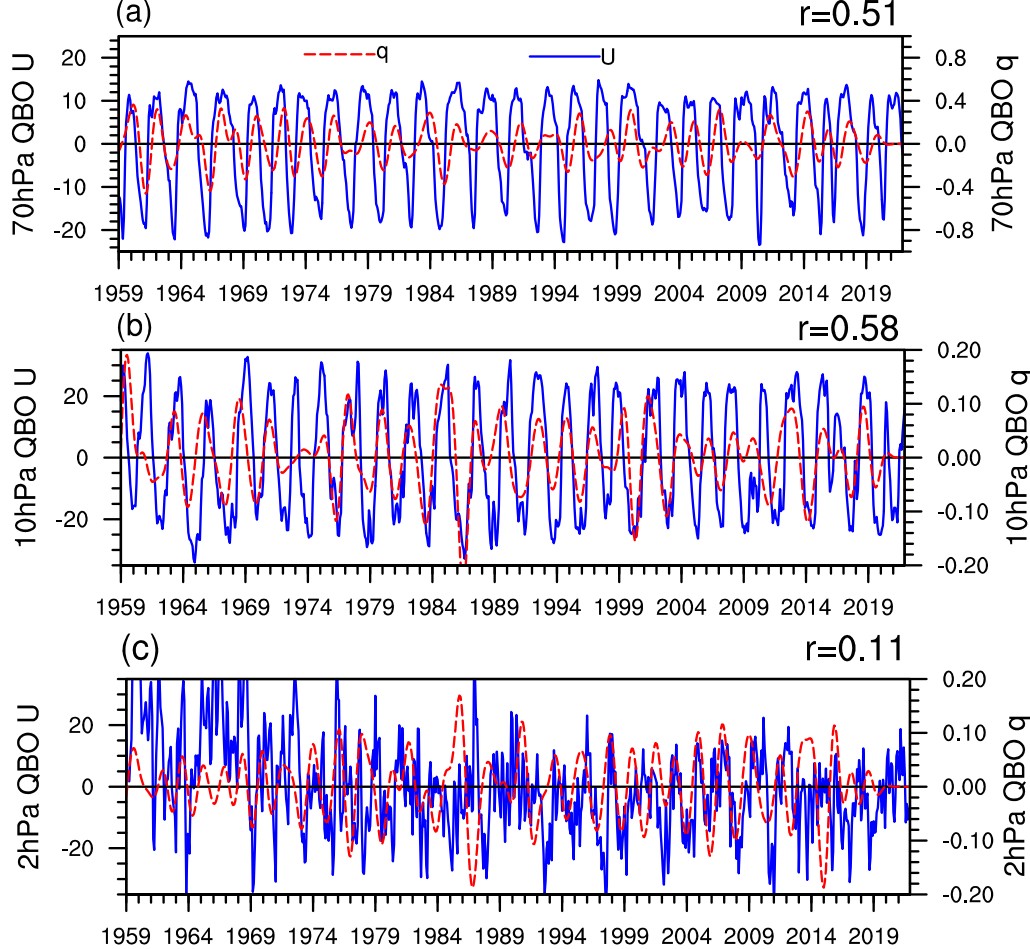

**Figure. 3 Zonal mean zonal wind anomalies (unit: m/s) and water vapor anomalies (unit: ppm) in the tropics (5°S–5°N) at (a) 70 hPa, (b) 10 hPa, and (c) 2 hPa. Water vapor anomalies are bandpass filtered to focus on periodicities of 15–60 months, and zonal wind anomalies are linearly detrended.**

## 4 Comparison of water vapor QBO between boreal winter and summer

*a. Spatial distribution of water vapor QBO*

Figure 4 shows the distribution of water vapor anomalies for different QBO phases in boreal winter and summer,

respectively. Under the QBO westerly phase in the northern summer, the water vapor anomaly is weak near the tropical



tropopause at 100 hPa, while a larger negative anomaly center is observed at 50 hPa with the amplitude of around -0.1 ppm. (Fig. 4a). Under the QBO easterly phase in the northern summer, the water vapor anomaly pattern is basically opposite to

175 that of the QBO westerly phase. Specifically, tropical water vapor near the tropopause is slightly reduced and at 30–70 hPa increased. This increase near 30–70hPa is displaced off the equator, towards the northern subtropics with the amplitude of 0.04 ppm (Fig. 4b). Because the distribution of water vapor anomalies in the westerly phase and the easterly phase in summer is almost opposite, the water vapor difference between westerly and easterly QBO phase is consistent with the distribution of water vapor anomalies in the westerly phase (Fig. 4c). It is also found that the tropopause in the tropics under

180 the QBO westerly phase rises higher than that under the easterly phase.

Under the QBO westerly phase in the northern winter, a deep and strong positive vapor anomaly band appears between 50-150 hPa in the tropics, with the central amplitude exceeding 0.1 ppm. Water vapor decreases in the middle and upper stratosphere around 5–30 hPa in the tropics, in contrast with the moistening in the upper stratosphere around 1–2 hPa (Fig. 4d). The distribution of water vapor anomalies in the QBO easterly phase in the northern winter is nearly reversed to that in

the QBO westerly phase. Water vapor around 50–150 hPa in the tropics and subtropics of both hemispheres decreases, with the maximum negative anomalies exceeding 0.1 ppm. Water vapor around 5–50 hPa in tropics increases, contrasted with the decrease around 2–5 hPa (Fig. 4e). The tropopause pressure of the QBO westerly phase in the tropics is about 5 hPa higher than that of the easterly phase.

In short, during both winter and summer, the influence of QBO on tropical stratospheric water vapor entry is nearly

190 symmetrical. Under the QBO westerly phase as an example, the distribution of tropical stratospheric water vapor displays a sandwich structure with positive, negative and positive water anomalies from the lower to upper layers. Further, the water vapor anomalies in the lower stratosphere during winter is stronger than during summer.

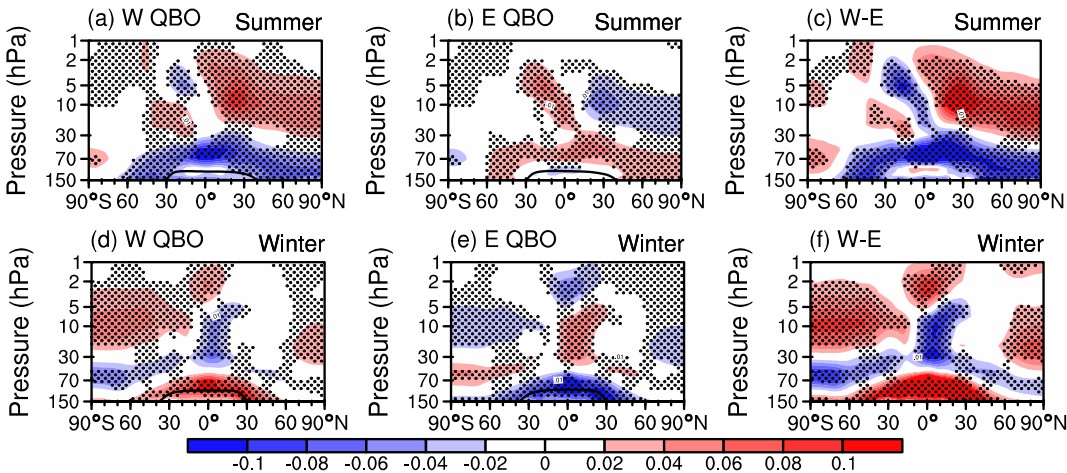

**Figure. 4 Water vapor anomalies under different QBO phases in the northern winter and summer (unit: ppm), respectively. Water**
**vapor anomalies were bandpass filtered with the periods from 15–60 months, and the black line marks the tropopause for QBO composite. (a) Composite of QBO westerly in northern summer. (b) Composite of QBO easterly in northern summer. (c) The difference between a and b.(d–f) As in a–c, but for northern winter.**




The primary source of stratospheric water vapor is tropical tropospheric water vapor entering the stratosphere. Motivated by Figure 2 which shows a pronounced QBO temperature signal at 100hPa, Figure 5 shows the tropical distribution of water

vapor anomalies at 100 hPa. During the QBO westerly phase in the northern summer, water vapor in the tropical Pacific increases, and the center of the positive anomaly magnitude exceeds 0.04 ppm (Fig. 5a). During the QBO easterly phase in the northern summer, water vapor decreases with the maximum center of 0.06 ppm in the equatorial middle Pacific region (Fig. 5b). These summer anomalies are much smaller than the winter anomalies. Namely, during the QBO westerly phase in the northern winter, tropical water vapor uniformly increases with the largest magnitude around 0.1 ppm (Fig. 5c), and the

dehydration during the QBO easterly phase in the northern winter also peaks at more than -0.1ppm (Fig. 5d). It is concluded that the amplitude of the water vapor response at 100hPa to the QBO forcing is nearly 2–3 times of that in summer.

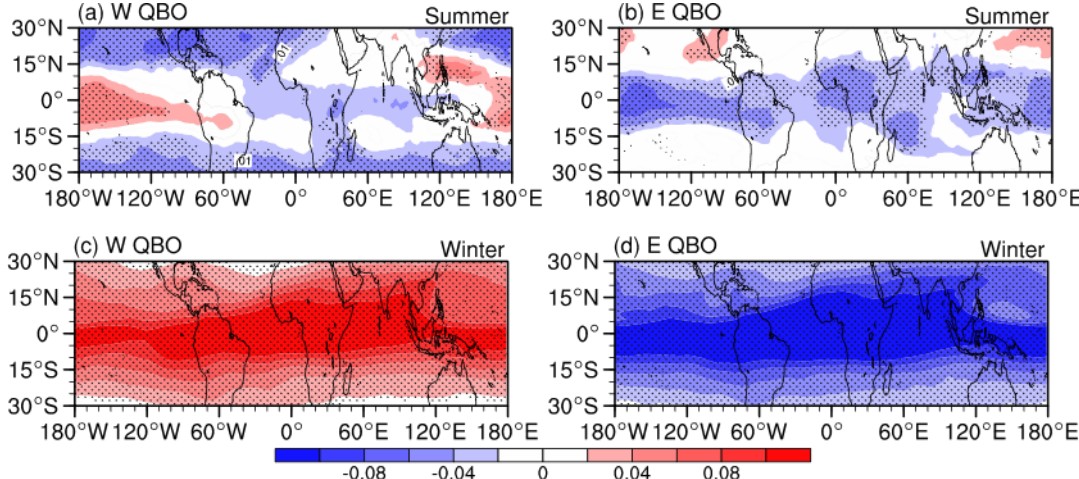

**Figure. 5 Distribution of water vapor anomalies at 100 hPa under different QBO phases in northern winter and summer (unit: ppm), respectively. Water vapor anomalies are bandpass filtered with the periods from 15–60 months. (a) Composite of QBO**
**westerly in northern summer. (b) Composite of QBO easterly in northern summer. (c, d) As in (a, b) but for northern winter.**

*b. Factors affecting the water vapor distribution*

Since the cold point temperature near the tropopause regulate entry water vapor (see the introduction), we now consider the temperature response to the QBO. Figure 6 shows the zonal mean temperature anomalies under different QBO phases in boreal summer and winter, respectively. It is found that the temperature anomalies in tropics–subtropics show a quadrupole

pattern in both boreal summer (Fig. 6a, b) and winter (Fig. 6c, d), although the temperature quadrupole is positioned differently.

Under the QBO westerly phase in boreal summer, the tropical lower stratosphere around 30–100 hPa warms, while the middle and upper stratosphere around 5–30 hPa cool, as expected from thermal wind balance. The temperature anomalies in the Southern Hemisphere subtropics are nearly opposite to those in the tropics, with cold anomalies in the lower stratosphere

and warm anomalies in the middle and upper stratosphere. A cold anomaly center with a magnitude of -1.5 K appears in the upper stratosphere over the Antarctic, corresponding to the strengthened polar vortex (Fig. 6a). Under the QBO easterly phase in boreal summer, the tropical lower stratosphere around 30–100 hPa cools, while the middle and high stratosphere





around 5–30 hPa warm. The temperature anomalies in the Southern Hemisphere subtropics are opposite to those in the tropics, with warm anomalies in the lower stratosphere and cold anomalies in the middle and upper stratosphere. A cold

anomaly center of ~4 K appears in the upper stratosphere over the Antarctic, corresponding to the weakening of the polar vortex (Fig. 6b).

Under the westerly QBO phase in boreal winter, the tropical lower stratosphere around 30–100 hPa is anomalously warm, and the middle stratosphere around 5–30 hPa is anomalously cold. The Northern Hemisphere subtropics are controlled by warm anomalies around 10–70 hPa and cold anomalies above. The upper stratosphere over the Arctic is anomalously warm,

and the middle and lower stratosphere is anomalously cold, corresponding to a strengthened Arctic polar vortex (Fig. 6c). Under the easterly phase in boreal winter, the lower stratosphere in the tropics is anomalously cold, and the middle and upper stratosphere around 10–30 hPa is anomalously warm. The Northern Hemisphere subtropics at around 10–30 hPa cool, and the stratopause is covered by warm anomalies. The Arctic stratopause is covered by cold anomalies, and the middle and lower stratosphere is dominated by warm anomalies, corresponding to the weakening of the polar vortex (Fig. 6d).

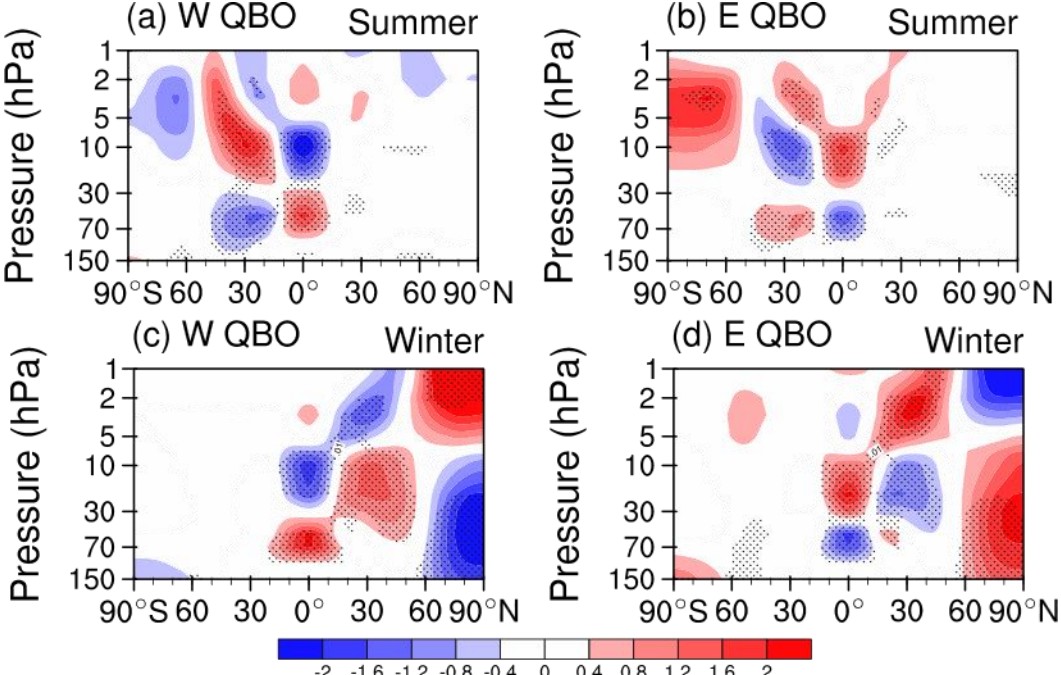

**Figure. 6 Zonal mean temperature anomalies under different QBO phases in northern winter and summer (unit: K), respectively. (a) Temperature anomalies during the QBO westerly phase in the northern summer. (b) Temperature anomalies during the QBO easterly phase in the northern summer. (c, d) As in a, b but for temperature anomalies for the northern winter. Dots mark the composite anomalies at the 95% confidence level.**

The Brewer-Dobson (BD) circulation in the stratosphere can directly affect the transport of water vapor from the tropical troposphere to the stratosphere (Holton et al. 1995; Butchart 2014; Keeble et al. 2021). The BD circulation also affects tropopause temperature in the tropics, which in turn affects water vapor entering the stratosphere (Abalos et al., 2021; Butchart, 2014; Hardiman et al., 2014). Figure 7 shows the BD circulation anomalies for different QBO phases in the



northern winter and summer. The QBO-related secondary circulation anomalies are only evident in the winter hemisphere
(i.e., the Southern Hemisphere in boreal summer and Northern Hemisphere in boreal winter), mainly due to enhanced active
planetary waves at midlatitudes in the winter hemisphere.

Under the QBO westerly phase in northern summer, anomalous upwelling appears in the lower stratosphere over the
Southern Hemisphere subtropics, accompanied with anomalous downwelling in the tropics. As a consequence, a clockwise
secondary circulation cell appears in the lower stratosphere. The residual circulation anomaly pattern in the middle and upper
stratosphere is reversed: anomalous rising motion over the tropics and sinking over the Southern Hemisphere subtropics
region (Fig. 7a). Under the easterly phase in boreal summer, the residual circulation anomalies show ascent in the tropical
lower stratosphere and descent in the lower stratosphere of the Southern Hemisphere subtropics, resulting in a
counterclockwise secondary circulation cell in the lower stratosphere. In the middle stratosphere, the residual circulation
anomalies show descent in the tropics and ascent in the subtropics, resulting in a clockwise circulation (Fig. 7b).

Under the westerly phase of the QBO in northern winter, the lower stratospheric residual circulation sinks in the tropics and
rises in the subtropics. The middle stratospheric residual circulation rises in the tropics and sinks in the Northern Hemisphere
subtropics (Fig. 7c). Under the QBO easterly phase in the northern winter, the lower stratosphere residual circulation rises in
the tropics and sinks in the Northern Hemisphere subtropics with the rising branch in the tropics stronger than the sinking
branch in the extratropics. The residual circulation in the middle stratosphere sinks in the tropics and rises in the Northern
Hemisphere subtropics. The descent over the Arctic is strengthened in winter, corresponding to weakening of the polar
vortex (Fig. 7d).

For all circumstances, the secondary circulation cell is consistent with the distribution of stratospheric temperature anomalies:
downwelling leads to warm and upwelling leads to cold via adiabatic heating/cooling. The anomalous secondary circulation
does not only lead to the advection of water vapor in the tropical stratosphere, but also affects the tropopause cold point
temperature.





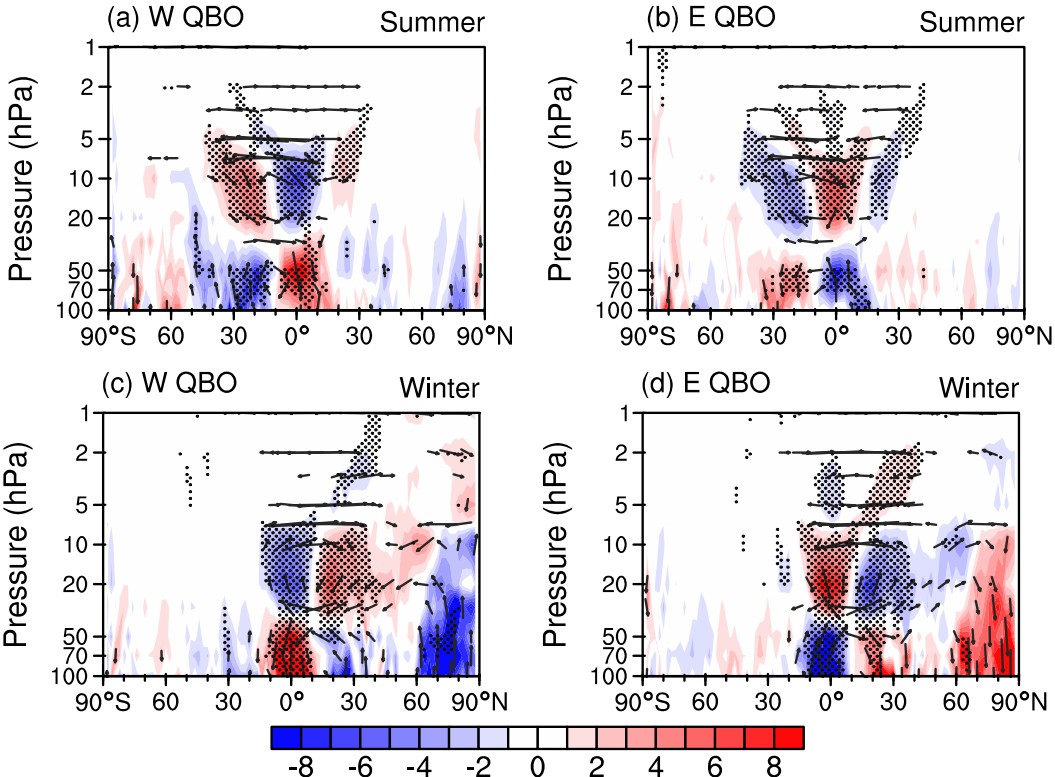

**Figure. 7 (a) Residual circulation anomalies during the QBO westerly phase in the northern summer. (b) Residual circulation anomalies during the QBO easterly phase in the northern summer. (c, d) As in a, b but for residual circulation anomalies for the northern winter. Dots mark the composite vertical residual velocity anomalies at the 95% confidence level. The shading is the vertical component of the residual velocities (units: $10^{-5}$ Pa/s).**

The temperature near the tropopause and in the lower stratosphere significantly affects the distribution of water vapor entering the stratosphere especially in the tropics (Garfinkel et al., 2013, 2021; Ueyama et al., 2016). Figure 8 shows the temperature and its anomalies at 100 hPa associated with the QBO variability in the northern winter and summer, respectively. Under the QBO westerly phase in the northern summer, the temperature anomalies in tropics are weak and insignificant (Fig. 8a). Under the QBO easterly phase in the northern summer, significant cold anomalies occur in the tropical Indian Ocean and broad part of tropical Pacific (Fig. 8b). Under the QBO westerly phase in the northern winter, warm anomalies appear in the tropical climatological coldest centers with the anomaly amplitude of ~0.45 K (Fig. 8c). while cold anomalies appear under the QBO easterly phase in the northern winter with comparable anomaly amplitude (Fig. 8d). Warm temperature anomalies over the equator are accompanied with anomalously more water vapor, while cold temperature anomalies are accompanied with anomalously less water vapor. This well explains the distribution of water vapor anomalies in the lower stratosphere in the tropics under both westerly and easterly phases of QBO. Comparing the temperature anomaly distribution in boreal winter versus summer, the coverage and amplitude of tropical temperature anomalies are very different: the lower stratospheric temperature variations associated with QBO in the tropics are much stronger in winter than in summer.





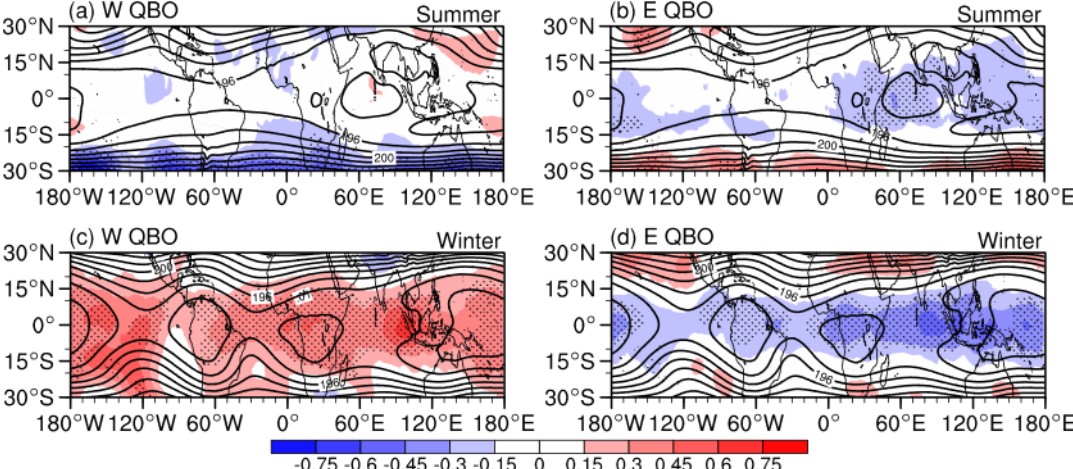

**Figure. 8 Temperature anomalies at 100 hPa under different QBO phases in northern winter and summer (shadings; unit: K), respectively. (a) Temperature anomalies during the QBO westerly phase in the northern summer. (b) Temperature anomalies during the QBO easterly phase in the northern summer. (c, d) As in a, b but for temperature anomalies for the northern winter. Contours show the temperature climatology in winter and summer (contour interval: 2), and dots mark the composite anomalies at the 95% confidence level.**

Previous studies have shown that deep convection produces warming in the upper troposphere and cooling near the cold point tropopause (Gettelman and Birner, 2007; Kim et al., 2018; Muhsin et al., 2018). Due to higher sea surface temperatures (SSTs) in the Indo-Pacific region, the convection over this region is relatively active while the tropopause cold point temperature is relatively low (Fueglistaler et al., 2009). This combination suggests that the QBO might be able to influence convection in this region. We now consider this possibility in Fig. 9. After using linear regression to eliminate the influence of ENSO on OLR, the OLR anomaly under QBO easterly and westerly phases was obtained. Under the QBO westerly phase in the northern summer, the outgoing longwave radiation (OLR) anomalies in the Indo-Pacific region are negative, indicating that the tropical deep convection is relatively active (Fig. 9a), which is beneficial to decrease the tropopause cold point temperature in tropics. Under the QBO easterly phase in the northern summer, positive OLR anomalies prevail in the Indo-Pacific region, indicating relatively inactive convection (Fig. 9b). Under the QBO westerly phase in the northern winter, positive OLR anomaly in the Indo-Pacific region indicates that convective activity is inhibited (Fig. 9c). Under the QBO easterly phase in the northern winter, OLR anomalies in the Indo-Pacific region are anomalously negative, indicating significant enhancement of convective activities (Fig. 9d). The OLR anomalies add extra warm anomalies during the QBO westerly phase in the northern winter and cold anomalies during the QBO easterly phase in the northern winter (Fig. 8c, d), which increases the temperature difference between the easterly and westerly QBO phases. This effect is relatively weak in the northern summer (Fig. 8a, b).



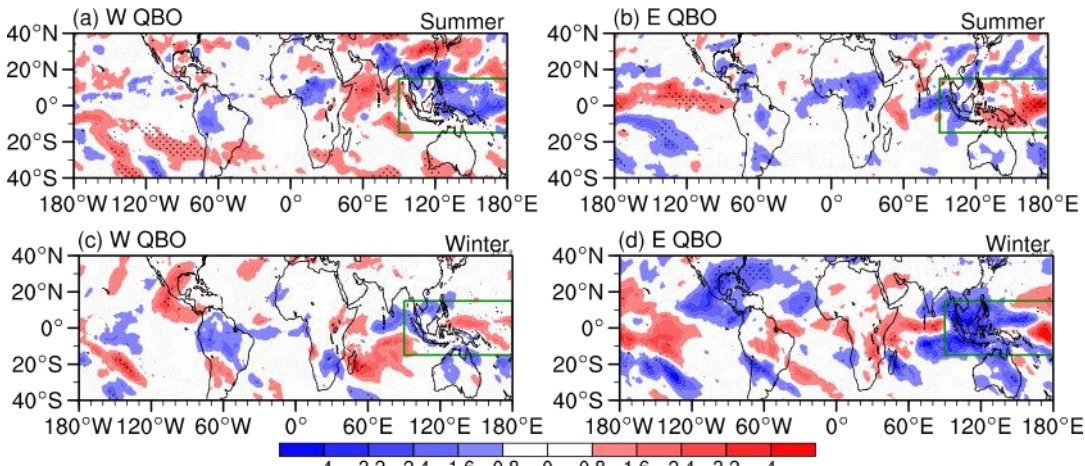

**Figure. 9 Outgoing longwave radiation (OLR) anomalies under different QBO phases in northern winter and summer (unit: W m⁻²), respectively. (a) OLR anomalies during the QBO westerly phase in the northern summer. (b) OLR anomalies during the QBO easterly phase in the northern summer. (c, d) As in a, b but for temperature anomalies for the northern winter. Dots mark the composite anomalies at the 95% confidence level.**

The transformed Eulerian-Mean (TEM) tracer continuity equation is used to quantify the balance of stratospheric water vapor associated with the QBO. Since the water vapor QBO is stronger in northern winter, the TEM diagnosis for the northern winter is shown in Fig. 10. Under the QBO westerly phase, the tropical water vapor shows a positive tendency in the lower stratosphere and a negative tendency in the middle stratosphere (Fig. 10a), explaining the water vapor anomalies (Fig. 4d). The first two terms on the right side of Eq. 1 are the mean transport of water vapor by the residual circulation (Fig. 10b). Namely, the residual circulation explains partially the water vapor variation in the tropical stratosphere. The tendency and mean transport of water vapor during the QBO easterly phase in the northern winter (Fig. 10e, f) are generally opposite to those during the QBO westerly phase. The residual item also exhibits negative water vapor anomalies in the tropical lower stratosphere (Fig. 10h).

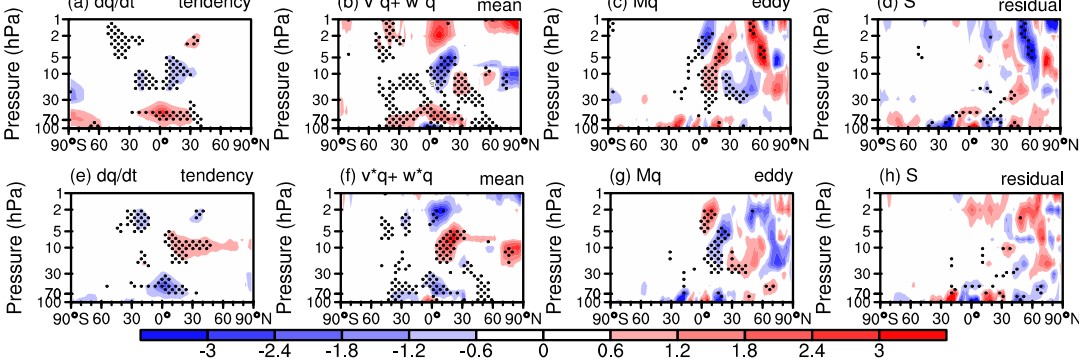

**Figure. 10 Diagnosis of the transformed Eulerian-Mean (TEM) tracer continuity equation in the northern winter. (a) Water vapor tendency during the QBO westerly phase (units: ppb day⁻¹). (b) The mean advection of water vapor during the QBO westerly phase (ppb day⁻¹). (c) The eddy transport of water vapor during the QBO westerly phase (ppb day⁻¹). (d) The residual term of water vapor during the QBO westerly phase (ppb day⁻¹). (e-h) The same as in a-d but for QBO easterly phase.**



## 5 Water vapor QBO in CMIP6 models

Since stratospheric water vapor has important climatic effects, evaluation of the simulated water vapor QBO by CMIP6 models is helpful in diagnosing how to improve the performance of the models (Keeble et al., 2021; Ziskin Ziv et al., 2022). Figure 11 shows the simulation of stratospheric water vapor QBO from 18 CMIP6 high-top models. Comparing the

simulated water vapor QBO from CMIP6 models with the ERA5 reanalysis data (Fig. 1), 11 models (ACCESS-CM2, AWI-CM-1-1-MR, CESM2-WACCM-FV2, CESM2-WACCM, GFDL-ESM4, HadGEM3-GC31-LL, HadGEM3-GC31-MM, MIROC6, MPI-ESM1-2-HR, MRI-ESM2.0, and UKESM1-0-LL) can effectively simulate the QBO variation of water vapor in the middle and lower stratosphere (10–100 hPa). The water vapor anomalies show continuous upward propagation from the lower to upper stratosphere in these models. The water vapor QBO in MIROC6 only propagates to 20 hPa, which is

relatively shallow compared to the ERA5 reanalysis. The water vapor QBO in the upper stratosphere around 1–5 hPa is simulated in eight models (AWI-CM-1-1-MR, CESM2-WACCM-FV2, CESM2-WACCM, CNRM-CM6-1, GFDL-ESM4, MIROC6, MPI-ESM1-2-HR, and UKESM1-0-LL). Differences in the height coverage of the water vapor QBO prevail among CMIP6 models.

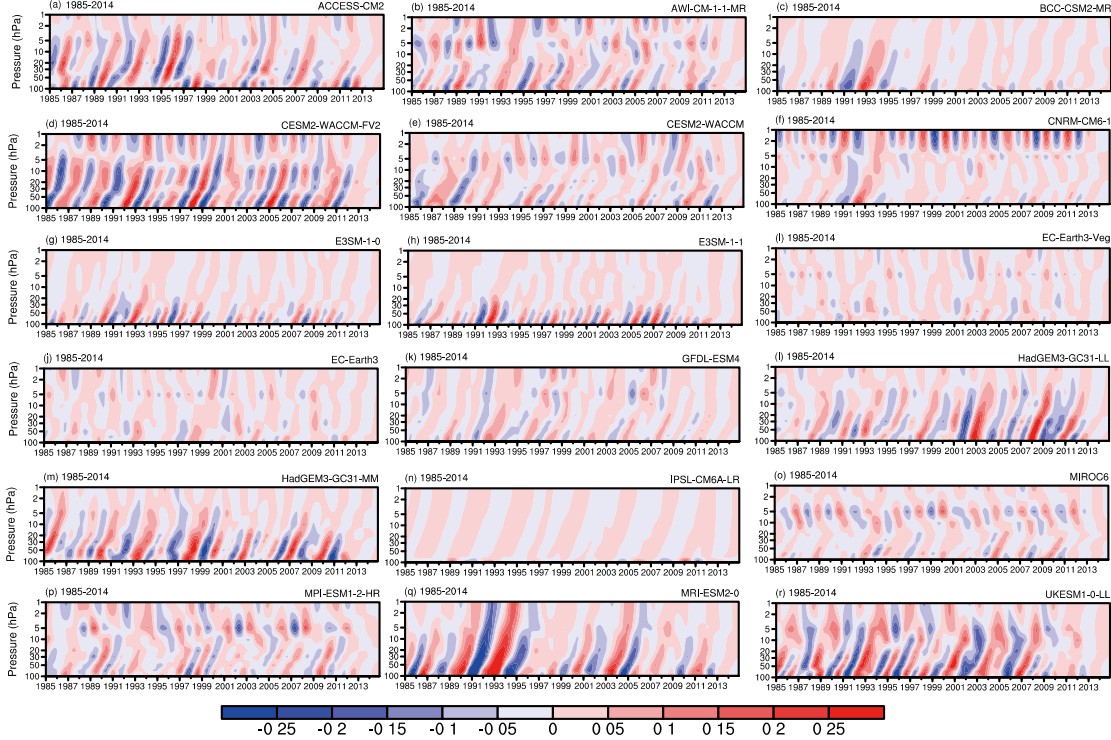

**Figure. 11 Historical simulation of water vapor QBO for (a–r) 18 CMIP6 models. Stratospheric water vapor anomalies in the tropics (5°S–5°N) during 1985–2014 have been bandpass filtered to focus on periodicities between 15 and 60 months (unit: ppm).**

The spatial pattern of the water vapor QBO in northern summer is shown in Fig. 12 for 18 CMIP6 models, respectively. In general, models that simulate a tape recorder effect in Fig. 11 also simulate a more realistic latitude vs. height response in



Fig. 12. The relative abilities of each model are quantified in Table 1, which calculates the spatial correlation coefficients of
water vapor at 100-1 hPa and 30°S - 30°N between ERA5 reanalysis and CMIP6 models. It is shown that the spatial
correlation coefficient between seven models and ERA5 exceeds 0.5 (AWI-CM-1-1-MR, CESM2-WACCM-FV2, E3SM-1-
1, EC-Earth3, MIROC6-CM2, MPI-ESM1-2-HR, MRI-ESM2-0), and they can simulate negative water vapor anomalies in
the tropical lower stratosphere under the QBO westerly phase and positive anomalies under the QBO easterly phase in
summer. Among the seven high-skill models, the water vapor anomalies in CESM2-WACCM-FV2 are the largest, while the
water vapor anomalies in EC-Earth3 are the weakest. The water vapor distribution simulated by CESM2-WACCM-FV2 and
MRI-ESM2-0 models under different QBO phases in northern summer agrees most with ERA5 (Table 1).

Figure 13 shows the simulation of water vapor anomalies under different QBO phases by 18 CMIP6 high-top models in
northern winter. Table 1 also calculates the spatial correlation coefficients of water vapor at 100-1hPa and 30°S-30°N
between ERA5 reanalysis and CMIP6 models. It can be observed that the water vapor QBO in boreal winter is more difficult
to reproduce than in summer for most models. Only five models (ACCESS-CM2, AWI-CM-1-1-MR, BCC-CSM2-MR,
HadGEM3-GC31-MM, IPSL-CM6A-LR) can simulate the positive water vapor anomalies in the tropical lower stratosphere
under the westerly QBO phase and the negative anomalies under the easterly QBO phase in winter. The spatial correlation
coefficient between BCC-CSM2-MR and ERA5 reanalysis is the highest in the tropics (0.82).



**Figure. 12 As in Figure. 4 but for composite water vapor anomalies under different QBO phases in the northern summer (unit: ppm), respectively, for 18 CMIP6 models.**







**Figure. 13 As in Figure. 12 but for composite water vapor anomalies under different QBO phases in the northern winter (unit: ppm), respectively, for 18 CMIP6 models.**



Most models struggle to simulate the contrast in water vapor between winter and summer. Most models only simulate the winter or summer water vapor QBO, and few models simulate the water vapor QBO pattern in both seasons. Only two models (AWI CM-1-1-MR and CESM2-WACCM-FV2) can simulate the seasonal contrast in water vapor distribution with the pattern correlation exceeding 0.5, although the general water vapor anomaly patterns show biases from the ERA5 reanalysis (Fig. 4).

**Table 1 Pattern correlation between ERA5 reanalysis and CMIP6 models for WQBO minus EQBO water vapor difference at 100-1hPa and 30°S-30°N.**

| CMIP6 model | Winter | Summer |
| --- | --- | --- |
| ACCESS-CM2 | 0.56 | 0.02 |
| AWI-CM-1-1-MR | 0.57 | 0.56 |
| BCC-CSM2-MR | 0.82 | 0.36 |
| CESM2-WACCM-FV2 | 0.29 | 0.87 |
| CESM2-WACCM | 0.08 | 0.34 |
| CNRM-CM6-1 | 0.07 | 0.43 |
| E3SM-1-0 | -0.61 | 0.39 |
| E3SM-1-1 | -0.08 | 0.50 |
| EC-Earth3-Veg | 0.19 | 0.45 |
| EC-Earth3 | 0.32 | 0.50 |
| GFDL-ESM4 | 0.45 | -0.18 |
| HadGEM3-GC31-LL | -0.42 | 0.09 |
| HadGEM3-GC31-MM | 0.67 | 0.20 |
| IPSL-CM6A-LR | 0.60 | 0.08 |
| MIROC6-CM2 | 0.36 | 0.59 |
| MPI-ESM1-2-HR | 0.32 | 0.52 |
| MRI-ESM2-0 | -0.35 | 0.86 |
| UKESM1-0-LL | 0.31 | 0.23 |
| MME | 0.76 | 0.87 |

Previous work has suggested that models with a stronger QBO in the lower stratosphere are better capable of simulating its effect on entry water vapor (Ziskin et al 2022), and we now consider this effect in these 18 models. Figure 14 shows the

scatter plots of QBO westerly phase minus easterly phase for the 50 hPa zonal wind index and 70 hPa water vapor anomalies in deep tropics among CMIP6 models. In summer, the 50 hPa QBO intensity simulated by the CMIP6 models is generally





weaker than ERA5, and the QBO index intensity is generally negatively correlated with the water vapor response (Fig. 14a). In winter, the positive correlation between the 50 hPa QBO intensity and the 70 hPa water vapor response is not present among CMIP6 models. The pattern correlation in Table 1 is used as a criterion to select high-skill models. Specifically, the

380 composites of 7 CMIP6 models (AWI-CM-1-1-MR, CESM2-WACCM-FV2, E3SM-1-1, EC-Earth3, MIROC6-CM2, MPI-ESM1-2-HR, MRI-ESM2-0) for summer QBO signals and 5 CMIP6 models (ACCESS-CM2, AWI-CM-1-1-MR, BCC-CSM2-MR, HadGEM3-GC31-MM, IPSL-CM6A-LR) for winter QBO signals are shown in Fig. 14. The spatial correlation coefficient between CMIP6 and ERA5 reanalysis reaches 0.87 for summer QBO signals. The water vapor anomalies above 70 hPa are basically consistent with the ERA5 reanalysis (Fig. 14a, b vs. Fig. 4a, b). The spatial correlation coefficient of

385 water vapor between CMIP6 and ERA5 reanalysis is 0.76 for winter QBO signals. While these models capture the spatial pattern of the water vapor anomalies, they struggle with the magnitude: the composite water vapor anomaly magnitude from CMIP6 models is only half of that in the ERA5 reanalysis.

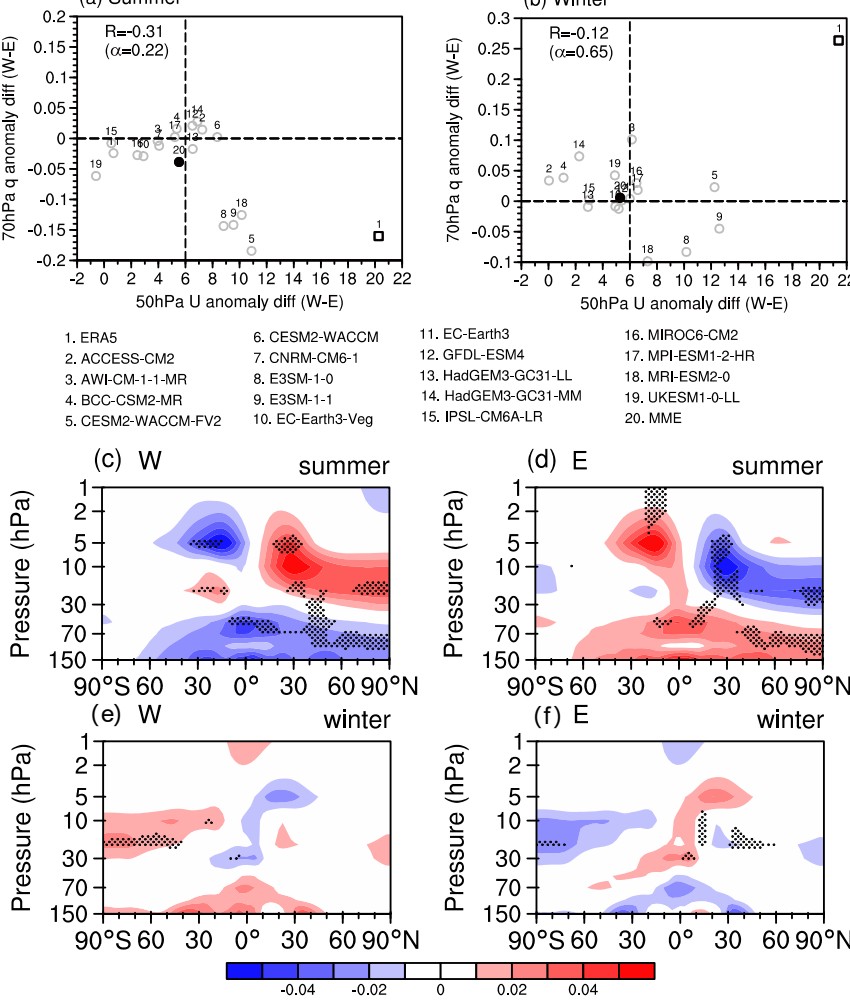



**Figure. 14 (a, b) Scatter plots of QBO westerly phase minus easterly phase for 50 hPa zonal wind anomalies and 70 hPa water vapor anomalies averaged between 5°S-5°N among models. (c-f) Composite of the 7 CMIP6 models in summer and 5 CMIP6 models in winter under different QBO phases.**

## 6 Summary and discussions

Based on ERA5 reanalysis and SWOOSH observations, this study investigates the water vapor QBO phenomenon in the stratosphere. The distribution of the stratospheric water vapor under different QBO phases in northern winter and summer is compared. The possible causes for the water vapor variability associated with the QBO are discussed, including the stratospheric circulation, temperature, and residual circulation variations. Simulations of the stratospheric water vapor QBO are also evaluated for 18 CMIP6 high-top models using the historical run data.

I. Previous studies have used SWOOSH and some climate models to analyze the tropical stratospheric water vapor entry associated with QBO (e.g., Ziskin et al., 2022). Here we focus on the QBO signals using a 15-60-month bandpass filtering of stratospheric water vapor in the deep tropics (averaged over 5°N-5°S latitude band), the tropical stratospheric water vapor presents obvious QBO variability, and the maximum water vapor QBO signals propagate regularly from the lowermost to middle and lower stratosphere around 10-100 hPa. The water vapor in the middle and lower stratosphere presents significant positive correlation with the QBO winds (correlation coefficient exceeding 0.5), which is consistent with the findings by Ziskin et al (2022). Namely, anomalous moistening typically occurs during the QBO westerly phase, while anomalous drying during the QBO easterly phase. Separate analysis on the water vapor QBO in the northern winter and summer reveals that the difference of water vapor distribution in the lower stratosphere between westerly and easterly QBO phases in summer is much smaller than in winter. Further, the water vapor QBO is also observed in the upper stratosphere around 2-5 hPa, although the water vapor anomalies are less significantly correlated with the zonal wind anomalies.

II. Difference in the secondary circulation associated with QBO between winter and summer is compared. In the northern summer, due to the dynamic effect of the QBO, two anomalous secondary circulation cells form in the lower and middle stratosphere over the tropics and the Southern Hemisphere subtropics region, respectively. The two secondary cells in the QBO westerly phase are opposite to those in the QBO easterly phase. The net effect is that the temperature anomalies manifest as a quadrupole distribution spanning the tropics and subtropics, with warm anomalies collocated with anomalous downwelling and cold anomalies collocated with anomalous upwelling. In the northern winter, the subtropical cell in the lower and middle stratosphere shifts to the Northern Hemisphere subtropics. These two secondary circulation cells are opposite under the easterly and westerly phases of QBO. The intensity of BD circulation anomalies in winter is obviously stronger than that in summer. Diagnosis of the transformed Eulerian-Mean (TEM) tracer continuity equation reveals that the mean advection term by the residual circulation associated with the QBO is the leading factor controlling the water vapor distribution in most of the stratosphere.





III.  The tropopause cold point temperature in the tropics affects the tropospheric water vapor entering the stratosphere (Garfinkel et al., 2013; Randel and Park, 2019). Consistent with previous work, the 100 hPa temperature in the tropics shows warm anomalies under the QBO westerly phase and cold anomalies under the easterly QBO phase. However, the intensity and coverage of tropical temperature anomalies in winter are significantly greater and broader than that in summer, which is consistent with the BD circulation anomalies being significantly stronger in winter than in summer. Additionally, OLR anomalies show opposite changes under both QBO phases in the northern winter and summer, respectively. The influence of OLR on the tropopause cold point temperature in summer is reversed with the tropical stratospheric low temperature anomalies, which further reduces the temperature anomaly magnitude around 100 hPa in summer. In northern winter, the effect of OLR on the cold point temperature is superimposed with the tropical lower stratospheric temperature anomaly, which amplifies the temperature difference between the easterly and westerly QBO phases.

IV.  Among the 18 CMIP6 high-top models that can produce the QBO spontaneously, 11 models can simulate upward propagation of the water vapor QBO from the lowermost to middle stratosphere around 10–100 hPa, though for all models the signal is too weak. Eight models simulate water vapor QBO in the upper stratosphere between 1-5 hPa, although divergence exists for the depth of the water vapor QBO among CMIP6 models. A model-by-model examination also reveals that the seasonal difference in the water vapor QBO can be reproduced by very few models, which challenge the model developers to well tune the simulation of the QBO itself and its climate effect.

The change of stratospheric water vapor can be traced back to the change of the cold point temperature at the bottom of the tropical stratosphere (Hardiman et al., 2015; Xia et al., 2019). The cold point temperature variability is a comprehensive effect from the stratospheric (from top to bottom) and tropospheric (from bottom to top) dynamics. In this study, the influence of stratospheric QBO on water vapor is considered, and the bottom-up influence from ENSO and sea surface temperature in the Indo-Pacific oceans is not considered. However, this study reveals the difference of the stratospheric water vapor QBO between winter and summer, and finds that BD circulation change related to QBO might be a mediator bridging the QBO and water vapor. It provides a new perspective to better understand the stratospheric water vapor QBO signals. As the dominant mode of the tropical stratosphere, a detailed analysis of the water vapor QBO signals is also a prerequisite to improve the performance of climate models for a better simulation of stratospheric variability and its role in subseasonal to seasonal forecasts.

## Acknowledgments

The work was supported by the National Natural Science Foundation of China (42361144843, 42322503, and 42175069), Israel Science Foundation (3065/23). NSFC and ISF are acknowledged for their funding. The authors thank ECMWF (https://cds.climate.copernicus.eu) for their providing the ERA5 reanalysis data. The SWOOSH data set version 2.7 is



obtained from https://csl.noaa.gov/groups/csl8/swoosh/. The CMIP6 data are provided by the WCRP (https://aims2.llnl.gov/search/cmip6).

**Autor contribution**

QL and JR designed the study. QL analyzed the data and wrote the manuscript. CS and CIG contributed to the discussion and revision of the paper.

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
