# Peer review of "Seasonality of the Quasi-biennial Oscillation signal in water vapor in the tropical stratosphere"

_EGUsphere, 2025_

## Author Comment (AC1)

**List of Responses**

**Responses to Farahnaz Khosrawi**

**Community Comments #1:**

Dear Farahnaz Khosrawi:

Thank you for your comments concerning our manuscript. Those comments are all valuable and very helpful for revising and improving our paper, as well as the important guiding significance for our research. We have studied the comments carefully and made corrections which we hope meet with approval.

This is an interesting study, but the authors need to motivate a bit more why it is interesting/important to investigate the seasonal cycle of the QBO. Just stating that this has not done before is, in my opinion, not enough.

Response: Thanks very much for your positive comments. We added why it is interesting/important to investigate the seasonal cycle of the QBO.

"Serva et al. (2022) found that there are seasonal differences in temperature and SWV in tropical regions. In the northern summer, the temperature at 100 hPa and the WV at 85 hPa reach their peaks, while in winter, they reach their lowest levels (Serva et al., 2022). The QBO is affected by the BD circulation, and it is stronger in northern winter than in summer (Butchart, 2014). Tegtmeier et al. (2020) found that the temperature amplitude of QBO was 2 K in February of northern winter and only 0.9 K in September of summer. Similar questions naturally arise: Does the amplitude of WV QBO also undergo a similar change? What are the differences between winter and summer? The research on the seasonal differences of WV QBO not only deepens the multi-time scale understanding of the stratospheric and tropospheric coupling, but also provides a scientific basis for cross-seasonal climate prediction. This study uses more samples based on the long time series of the QBO signal in SWV and discusses the differences in SWV distribution between different QBO phases and between different seasons. Possible causes of those differences are diagnosed, and the performance of climate models in capturing the QBO signal in WV is also evaluated (Ye et al., 2018; Ziskin et al., 2022)." (L70-81)

Following the description of results was for me quite difficult since there were on one hand too many figures and on the other hand these were not well explained. Starting with Figure 2 it is not stated which data set has been used for the analysis and thus which data set is shown on the figures. I could also find no statement on this in the main text. Are you using here a multi-model mean? Or the SWOOSH data? Or is the analysis based on the reanalysis data?

Response: Thank you for your criticisms. Following Figure 2, we added more discussions this time. The subsequent analysis is based on ERA5 reanalysis.

 "By comparison, it is found that although there are some differences in the SWV QBO between the SWOOSH satellite data and the ERA5 reanalysis data, the ERA5 reanalysis data reproduce the distribution pattern of WV propagation from the lower stratosphere to the upper stratosphere below 10 hPa. Therefore, the long-term data from ERA5 reanalysis can still be used to diagnose the influence and dynamics of WV in the middle and lower stratosphere below 10 hPa. Our subsequent analysis mainly uses ERA5 reanalysis data." (L175-179)

It should be made more clear which data set has been used for what in your analysis. Also a clear statement/discussion on the uncertainty of these data sets is missing. Response: Thank you for your suggestion. We have added the comparison of water vapor between the SWOOSH satellite observation data and the ERA5 reanalysis in the data and methods, and introduced the uncertainty of the data.

- "This dataset is a combination of different data sources, including SAGE-II/III/ISS, UARS HALOE, UARS MLS, Aura MLS, ACE-FTS, and OMPS-LP (Davis et al., 2016). The SWOOSH data provide monthly averages, standard deviation, number of observations and average uncertainty on the pressure grid measured by each satellite instrument. SWOOSH also includes combined (multi-instrument) products based on the weighted average of available measurement values. A key aspect of the merged product is that the source records are homogenized to account for inter-satellite biases and to minimize artificial jumps in the record, producing a long-term data record. Since the UARS HALOE observation data began on 19 October 1991, we have used SWOOSH data from 1992 onwards." (L97-103)
- "In contrast to the troposphere, the WV content within the stratosphere is extremely low....." (L104-)

**Specific comments:**

Introduction: Here it should also be discussed that many models have/had problems in simulating QBO and that for that considering waves is important (e.g. Giorgetta et al. 2006). How has this overcome in the CMIP6 model simulations? Which effort has been made so that the models are able to simulate a QBO.

Response: Thanks for your suggestion. We have added this content.

"However, it remains a challenge to simulate the QBO in general circulation models (GCMs), with only a few GCMs being able to reproduce it. The waves need to be correctly represented to simulate a realistic QBO. Many GCMs still cannot simulate a realistic spectrum of tropical waves because of their low resolution and their deficiencies in the parameterization of small-scale gravity waves forcing (Ricciardulli and Garcia, 2000; Lott et al., 2014). Studies have suggested that an adequately fine vertical resolution (vertical grid spacing of ~500–700 m) of the troposphere and lower stratosphere is also necessary to simulate the QBO due to the forcing of some resolved waves with small vertical wavelength and the need to capture the wind shear (Richter et al., 2014b; Geller et al., 2016). In CMIP5, only five models could generate the QBO internally (Butchart et al., 2018). In CMIP6, at least 15 models now able to simulate realistic QBO-like behavior during the historical period (Richter et al., 2020)." (L49-57)

P3, L65: Here you should provide a motivation why investigating the seasonal cycle of the QBO is important/of interest.

Response: The research significance has been added to the Introduction.

"Serva et al. (2022) found that there are seasonal differences in temperature and SWV in tropical regions. In the northern summer, the temperature at 100 hPa and the WV at 85 hPa reach their peaks, while in winter, they reach their lowest levels (Serva et al., 2022). The QBO is affected by the BD circulation, and it is stronger in northern winter than in summer (Butchart, 2014). Tegtmeier et al. (2020) found that the temperature amplitude of QBO was 2 K in February of northern winter and only 0.9 K in September of summer. Similar questions naturally arise: Does the amplitude of WV QBO also undergo a similar change? What are the differences between winter and summer? The research on the seasonal differences of WV QBO not only deepens the multi-time scale understanding of the stratospheric and tropospheric coupling, but also provides a scientific basis for cross-seasonal climate prediction. This study uses more samples based on the long time series of the QBO signal in SWV and discusses the differences in SWV distribution between different QBO phases and between different seasons. Possible causes of those differences are diagnosed, and the performance of climate models in capturing the QBO signal in WV is also evaluated (Ye et al., 2018; Ziskin et al., 2022)." (L70-81)

P3, L83-87: Add a few more sentence on the SWOOSH data set itself and the quality of this data set. Have all e.g. biases been removed? How have the satellite data sets been merged and what is the advantage of using this merged data set instead of using one or several satellite data sets separately?

Response: Added.

"SWOOSH also includes combined (multi-instrument) products based on the weighted average of available measurement values. A key aspect of the merged product is that the source records are homogenized to account for inter-satellite biases and to minimize artificial jumps in the record, producing a long-term data record." (L100-102)

P5, L137: What do these differences mean for your study? Which data set is more realistic? The SWOOSH QBO or the reanalysis QBO? What are the known uncertainties of these data sets?

Response: Here, water vapor below 10 hPa is displayed, and detailed descriptions of ERA5 and SWOOSH are provided.

• "In terms of data and methods, we compared ERA5 reanalysis with SWOOSH satellite monitoring data and found that ERA5 reanalysis data could reproduce the distribution pattern of SWV (Fig. 1). ERA5 reanalysis can well display the QBO signal of SWV below 10 hPa......" (L164-)

"Compared to the SWOOSH satellite observation data, the ERA5 reanalysis data provides a longer time span, which provides more samples for revealing the effect of QBO on SWV. There remains uncertainty regarding the performance of ERA5 reanalysis data in depicting SWV......" (L104-)

P5, L124: In Figure 2 the QBO from ERA5 and SWOOSH is shown but then the QBO is analysed in detail but without stating which data set has been used.

Response: In the newly revised manuscript, the water vapor QBO shown by ERA5 and SWOOSH were analyzed respectively.

• "Figure 2 shows the evolution of SWV anomalies in the tropics over the past 60 years for the ERA5 reanalysis and 30 years for the SWOOSH data." (L168-169)

Figure 2-10: Which data set has been analysed?

Response: Figure 2-10 used ERA5 reanalysis data.

· "Our subsequent analysis mainly uses ERA5 reanalysis data....." (L178-)

P7, Figure 3: In the figure it is written "q". If the specific humidity is shown here this should be clearly stated or if you have calculated from q the water vapour mixing ratio then you should write in the figure legend and on the axis "H2O".

Response: Changed to H2O in Fig.4. (L210)

P11, L263: Leads to cold "temperatures"? Please be more clear.

Response: Added:

 "Namely, when an air mass descends, it is compressed, its volume decreases, its internal energy increases, and its temperature rises. This phenomenon is called adiabatic heating. Conversely, during ascent, adiabatic cooling occurs." (L317-319)

Figures: 14 figures are too much. I would suggest to put some of the figures in an appendix and focus in the main text of the manuscript on the most important figures. Response: Changed. Thanks for your suggestion. We have moved some figures from the main manuscript to the supplementary materials, leaving 11 figures in the main text.

**Technical corrections:**

P7, L162: shows weak -> shows a weak

Response: Corrected.

P7, Figure 3 caption: "Figure 3" should be "Figure 3." Note, this needs to be corrected also for all other figures, too.

Response: All the figure captions have been changed.

P8, L179: un der -> under

Response: It has been removed in the latest revised version.

P9, L206: 100hPa -> 100 hPa

Response: Changed. (L229)

P10, L223: warm -> warms Response: Changed. (246)

P13, L299: in tropics -> in the tropics

Response: It has been removed in the latest revised version.

**References:**

Giorgetta, M. A., Manzini, E., Roeckner, E., Esch, M., and Bengtsson, L.: Climatology and forcing of the quasi-biennial oscillation in the MAECHAM5 model, J. Clim., 19, 3882–3901, 2006.

Response: Learned and cited this reference.

---

## Author Comment (AC2)

**Responses to Reviewer #2**

**Reviewer #2**

I recommend rejection of this manuscript as it suffers from two fundamental flaws:

- 1. ERA5 Stratospheric Water Vapor (SWV) is the central dataset in this manuscript. It is studied in sections 3 and 4 and it is used as reference for CMIP6 evaluation (section 5). The section is a first section of the ERA5 contains the section of the section of
- 5). The manuscript fails to consider the nature of the ERA5 reanalysis of water vapour, i.e. an optimal representation between model state, in-situ humidity observations in the troposphere and satellite radiance observations which are sensitive to humidity only in the troposphere. This appears clearly in Hersbach et al. (2020) where section 5 provides an exhaustive description of the assimilated observations.

Response: Thank you for your criticism. We agree that ERA5 SWV might have some biases. However, we still believe that the bias in ERA5 is smaller than that in models relative to observations.

To well address your concern, we assess the quality of the ERA5 SWV with the SWOOSH as the baseline. It is revealed that the ERA5 quality is not so bad as imagined. The assessment of ERA5 SWV is provided in Figure 1, and our reviewers might find that the SWV in ERA5 and SWOOSH is highly similar especially below 10 hPa.

For your concern, we also add more discussion of the ERA5 assimilation of WV.

"In contrast to the troposphere, the WV content within the stratosphere is extremely low. Compared to the SWOOSH satellite observation data, the ERA5 reanalysis data provides a longer time span, which provides more samples for revealing the effect of QBO on SWV. There remains uncertainty regarding the performance of ERA5 reanalysis data in depicting SWV. In the ERA5 reanalysis, WV mainly assimilates in-situ humidity observations in the troposphere and satellite radiation observations that are only sensitive to humidity in the troposphere. Therefore, the SWV in ERA5 reanalysis data is a GCCM output with specified dynamics (Hersbach et al., 2020). From 2000 to 2006, ERA5 showed cold deviation in the lower stratosphere. The global average temperature of the stratosphere and tropospheric apex corrected by ERA5.1 was better than that of ERA5 (Simmons et al., 2020). Previous studies have shown a wet bias in the tropical tropopause in the ERA5 reanalysis data (Kruger et al., 2022). Some studies also found that the content of SWV in ERA5 was superior to that of ERA-Interim (Wang et al., 2020). We use SWOOSH satellite monitoring data to validate the applicability and uncertainty of ERA5 reanalysis data in SWV. Figure 1 shows the evolution of the annual mean, summer mean, and winter mean of the WV mixing ratio in the troposphere during SWOOSH satellite data and ERA5 reanalysis from 1992 to 2019. Compared to SWOOSH satellite data, ERA5 can better display the distribution pattern of SWV, and the WV content is basically consistent. The WV content in ERA5 has a 0.3 ppm moisture deviation at the bottom of the tropical stratosphere, which is reflected in the annual mean, summer mean and winter mean, consistent with previous analyses (Kruger et al., 2022). However, at the top of the stratosphere, the WV content in ERA5 is all relatively low." (L104-119)

We hope our reviewer can understand that the ERA5 really provides more QBO samples than other datasets before we have enough observed WV time series. Thank you for your criticism again.

While satellite retrievals from limb-scanning instruments are assimilated after 2002 in the case of ozone (i.e. MIPAS and Aura-MLS), no similar dataset is assimilated for water vapor. SWV has a negligible impact on satellite observations of microwave and infrared radiances because they are observed in a nadir-looking geometry. Hence ERA5 SWV is not influenced by these observations, is fundamentally nothing more than the output of a GCCM with Specified Dynamics, and is thus far from a "true" dataset. This is also indicated by Hersbach et al. in their Fig. 12, where analysis increments of humidity are not shown above 300 hPa while analysis increments of temperature, zonal wind and ozone are also shown at 50 hPa and 3 hPa; and in the discussion of their Fig. 19 (SWV above the South Pole) where it is specifically written that "no humidity observations are assimilated at this level" (850 K isentropic surface i.e. midstratosphere).

Response: We agree that ERA5 reanalysis can not represent the real world. However, the ERA5 reanalysis provides more QBO samples. To well address your concern, we give a direct comparison between ERA5 and SWOOSH for the zonal mean WV from 1000 hPa to 1 hPa. It is revealed that the WV above 10 hPa diverges between two datasets. However, the zonal mean WV pattern below 10 hPa shows somewhat similarity between two datasets. Also see the response above.

We also compare the timeseries of the tropical WV from 1990-2020 in Figure 2. We noticed that the WV anomalies above 30 hPa are weaker in ERA5 than in SWOOSH. We mainly focus on the WV below 30 hPa (especially in Figure 5).

We also give more discussion on the comparison between both datasets.

• "By comparison, it is found that although there are some differences in the SWV QBO between the SWOOSH satellite data and the ERA5 reanalysis data, the ERA5 reanalysis data reproduce the distribution pattern of WV propagation from the lower stratosphere to the upper stratosphere below 10 hPa. Therefore, the long-term data from ERA5 reanalysis can still be used to diagnose the influence and dynamics of WV in the middle and lower stratosphere below 10 hPa. Our subsequent analysis mainly uses ERA5 reanalysis data." (L175-179)

The accuracy of ERA5 in the stratosphere (and its 2000-2006 correction ERA 5.1) are summarized in section 7 of Hersbach et al. (2020 – see especially their Fig. 15) and extensively discussed by Simmons et al. (2020, not cited in the manuscript). Yet such uncertainties are not considered in this manuscript, which also fails to mention the well-known moist bias in ERA5 at the tropical tropopause (Krüger et al., 2022). An earlier

study of stratospheric water vapor in ERA5 did conclude that SWV is better represented in ERA5 than in ERA-Interim (Wang et al., 2020) but this earlier study of the QBO in ERA5 SWV is not cited either.

Response: All those references are mentioned and cited this time. We list all the revisions for your reference.

- "From 2000 to 2006, ERA5 showed cold deviation in the lower stratosphere. The global average temperature of the stratosphere and tropospheric apex corrected by ERA5.1 was better than that of ERA5 (Simmons et al., 2020). Previous studies have shown a wet bias in the tropical tropopause in the ERA5 reanalysis data (Kruger et al., 2022). Some studies also found that the content of SWV in ERA5 was superior to that of ERA-Interim (Wang et al., 2020)." (L109-113)
- "Compared to SWOOSH satellite data, ERA5 can better display the distribution pattern of SWV, and the WV content is basically consistent. The WV content in ERA5 has a 0.3 ppm moisture deviation at the bottom of the tropical stratosphere, which is reflected in the annual mean, summer mean and winter mean, consistent with previous analyses (Krüger et al., 2022)." (L115-118)

The confusion between ERA5 SWV and observed SWV could have been partially overcome by using the SWOOSH dataset, which is entirely based on observations. Yet SWOOSH is only used for Fig.1. After a very superficial comparison between ERA5 and SWOOSH SWV (line 134), the authors seem to believe that ERA5 assimilated HALOE SWV data (lines 135-136) while this is not true.

Response: Thanks very much for your suggestions. Your suggestions have enabled us to have a deeper understanding of the ERA5 reanalysis data of water vapor. We have provided detailed explanations and discussions in the data and methods sections. (L104-124)

The comparison of water vapor in ERA5 reanalysis data and SWOOSH data was also made in Figures 1, 2 and Figure S1. (L164-179)

2. This manuscript attempts to study the seasonality of the QBO signal in water vapor by comparing composite of **anomalies** in "boreal summer" and "boreal winter" (which I understand as JJA and DJF, respectively). While the methodology section severely lacks details, one can still read (line 99) that "The anomaly refers to the deviation of the monthly data from the monthly climatology...". Since these anomalies are the signal **after removal of the seasonal cycle**, it makes no sense to study the differences between their composites for DJF and JJA. Yet this method is at the core of the manuscript, as shown by Figs 4 to 10 and 12 to 14. Furthermore, section 2 states that a "Butterworth first-order bandpass filter was used to extract the water vapor variations at the period of 15-60 months". How can the seasonality of the dataset be preserved after the application of such a filter? The methodology used here would have been greatly clarified (and probably invalidated) by showing time series of SWV prior to calculation of anomalies, and also prior to the application of the bandpass filter.

Response: We understand your concern. QBO has a spectrum maximum far from the annual cycle. It is more reasonable to remove the annual cycle of focused variables

before we study the composite QBO signals. The annual cycle is naturally related to the revolution of the Earth, which is not the focus of our study. In contrast, the QBO has a period of around 28 months. We use a wider period spectrum to keep the QBO signals (15-60 months).

In short, we use the anomaly fields to study with an expectation of removing the interference of the annual cycle.

Compare with the methodology used and explain in much better detail by Wang et al. (2020), where seasonal cycle in ERA5 SWV is clearly shown by **anomalies from the climatological annual mean** (fig. 2) while the QBO cycle in ERA5 SWV is shown by **anomalies relative to the monthly climatology** (fig. 3).

Response: We read the paper by Want et al. (2020). Anomalies should be the deviation from the annual cycle. Please also see the responses below.

The large differences found here between DJF and JJA anomalies are thus quite difficult to interpret. They could be due to inconsistent periods for the removal of the monthly climatologies. In other words: the period used for the climatological seasonal cycle is 1960-2020 (line 98) but are the "composite" SWV anomalies shown from Fig. 4 onwards also computed for that period?

Response: Thanks for your suggestion. As you said, we used monthly climatology to calculate the anomalies so as to show the water vapor QBO characteristics. Wang et al. also showed the water vapor QBO characteristics in the monthly anomaly (Fig.v3). We introduced the calculation method of monthly anomalies in the data and methods.

• "The climatology of a variable is calculated as long-term monthly average over the time period from 1960 to 2020. The anomaly refers to the deviation of the monthly data from the monthly climatology with the trend removed for each calendar month." (L136-137)

Figure 9 in Want et al. shows the water vapor anomalies in northern winter and summer. By extracting several points, such as 1985, 1990, 2005, etc., it can be clearly found that the water vapor anomaly in winter is stronger than that in summer. We copied their figure below for your reference.

**Figure R1.** Time series of mean tropical SWV anomaly (15°S–15°N mean at 15–20 km) in (a) DJF and (b) JJA. Data is from ERA5. (Wang et al.'s Figure 9)

The Butterworth filtering is also shown in Wang et al.'s Figure 3 is copied below for your reference. They show that the stratospheric water vapor presents obvious periodic changes of 8-10 years. Therefore, we used the filtering method to extract more significant QBO signals (Fig.2, Fig.S1). You are concerned that the filtering might filter out the seasonal signals, so we showed the original water vapor anomaly in Figure S2, S3.

**Figure R2.** Tropical SWV anomalies (15°S-15°N mean) relative to the monthly climatology using (a, b) ERA5 and (c) SDI MIM. The panels (b, c) are based on the 15-20 km mean. Unit: 0.1 ppm (Wang et al.'s Figure 3)

Incorporating feedback from other reviewers and considering the lagged influence of the QBO on lower stratospheric water vapor, we found that the impact of the 30 hPa QBO index on 100 hPa water vapor peaks at a six-month lag. Although the lag effect remains stronger in winter than in summer, the response sign is identical between two seasons now. (Figs. 4, 5). (L220-230)

**Other comments**

In case the manuscript undergoes major revisions, the following issues should be addressed as well:

1. The abstract should better introduce the key concepts, i.e. the QBO and "the seasonal difference in the water vapor QBO".

Response: The abstract has been rewritten to clearly distinguish between the QBO and the water vapor QBO. (L10-23)

2. The introduction itself is not written in a rigorous manner:

The very strong statement of the first sentence ("Water vapor is the dominant greenhouse gas in the atmosphere") is not clearly supported by the two provided references (Dessler et al., 2013; Solomon et al., 2010) which are about the feedback between SWV and tropospheric climate.

Response: Revised.

• "The feedback of water vapor (WV) has an impact on global temperature changes (Held and Soden, 2000; Dessler et al., 2013; Solomon et al., 2010)....." (L25-26)

The modulation of chemical processes by SWV (line 31) is actually a weak feedback, as explained by Wohltmann et al. (2024). Tian et al. (2023) does not address specifically this statement; I was not able to check Tian et al. (2009) as this reference is incomplete. Response: Thanks for your suggestion. This sentecen was revised.

• ".....and the increase in SWV, for example, has a slight impact on ozone depletion (Tian et al., 2009, 2023; Wohltmann et al., 2024)." (L29-30)

Admittedly, "the seasonality of the water vapor QBO signal has been seldom studied" (line 65). But why is it interesting to study this seasonality? The introduction fails in its primary aim.

Response: Thanks very much for your positive comments. We added the motivation of this study.

- "Serva et al. (2022) found that there are seasonal differences in temperature and SWV in tropical regions. In the northern summer, the temperature at 100 hPa and the WV at 85 hPa reach their peaks, while in winter, they reach their lowest levels (Serva et al., 2022)....." (L70-)
- 3. Many more details are necessary about the datasets (section 2a). What is their time resolution at download time (i.e. prior to derivation of the anomalies): hourly, daily, monthly? Has ERA-5 been downloaded on model levels or (much less accurate) on pressure levels? What are the uncertainties of SWV in SWOOSH (see comment 1 for

ERA5)? Two generic papers are cited for CMIP6, but what are the specific references about the CMIP6 historical simulations? Could a reference be found about the overall quality of SWV in these simulations?

Response: All those details are added.

- "A horizontal resolution of 1° (latitude) × 1° (longitude) at 37 pressure levels from 1000 to 1 hPa in the vertical direction was collected for this study. The variables used include zonal and meridional wind, specific humidity, and air temperature on pressure levels." (L92-94)
- "In contrast to the troposphere, the WV content within the stratosphere is extremely low. Compared to the SWOOSH satellite observation data, the ERA5 reanalysis data provides a longer time span, which provides more samples for revealing the effect of QBO on SWV." (L104-)
- "Previously, some studies found that the CMIP6 models underestimate the WV content at the bottom of the tropical stratosphere (Keeble et al., 2021; Ziskin et al., 2022)." (L133-134)
- 4. The base climatology is calculated for the period 1960-2020. Does it make sense two very different periods of ERA5 (compare fig. 1a with fig. 1b) to compute its base climatology? See also comment 2. The period 1960-2020 presumably applies only to ERA5, as the CMIP6 simulations end in 2014. Does it make sense to compare anomalies for different periods?

Response: Thank you for your suggestion. Below 10 hPa, the water vapor QBO signal of ERA5 is relatively consistent in the two periods. We have modified Figure 2 to visually display the water vapor below 10 hPa.

Actually, the filtering does not depend on the climatology.

- 5. Line 104-105 define criteria of +/- 5 m/s for QBO "events". Are these the criteria used to build the W QBO and E QBO composites? Please clarify. Response: Clarified.
- "QBO events are selected when the QBO index is greater than 5 (less than -5) m/s to build the westerly QBO (easterly QBO) composites, following previous studies (e.g., Rao et al. 2020a, 2020b)." (L142-144)
- 6. All water vapor anomalies are shown in ppm units. Does this refer to water vapor volume mixing ratio (a.k.a. mole fraction) or mass mixing ratio?

  Response: All figure captions modified to "(mass mixing ratio, units: ppm)".
- 7. Figure 2 shows only two contour lines for the zonal wind, with no labels. Yet lines 142-147 discuss the numerical values, which are impossible to read from the figure. Response: Figure 3 caption added. "The contours are shown at  $\pm 15$  m/s and  $\pm 30$  m/s." (L193)

8. Similarly for fig.4: lines 179-180 and 187-188 discuss a difference in tropopause pressure between the W QBO and the E QBO, but this can not be seen on the figure. BTW, how is defined the tropopause here? Are you using a thermal, dynamical or SWV-based definition?

Response: Thanks for your suggestion. We have removed the discussion regarding the tropopause height differences. The original Fig. 4 has been moved to the supplementary materials.

9. What is the meaning of the dotted regions on Fig. 4?

Response: Moved to Fig. S2.

- "Dots denote statistical significance at the 95% confidence level based on Student t-test."
- 10. Figures 12 is not explained at all, and discussed very succinctly with figure 13 and Table 1 (lines 344-359). This should be expanded.

Response: Thanks for your suggestion. In response to your feedback and in consideration of the other reviewers' comments, the original Figures 12, 13 have been removed to reduce the figures.

**Additional references**

Krüger, K., Schäfler, A., Wirth, M., Weissmann, M., & Craig, G. C. (2022). Vertical structure of the lower-stratospheric moist bias in the ERA5 reanalysis and its connection to mixing processes. *Atmospheric Chemistry and Physics*, 22(23), 15559-15577.

Simmons, A., Soci, C., Nicolas, J., Bell, B., Berrisford, P., Dragani, R., ... & Schepers, D. (2020). Global stratospheric temperature bias and other stratospheric aspects of ERA5 and ERA5. 1.

Wang T, Zhang Q, Hannachi A, Hirooka T, Hegglin MI. Tropical water vapour in the lower stratosphere and its relationship to tropical/extratropical dynamical processes in ERA5. *Q J R Meteorol Soc.* 2020; 146: 2432–2449. https://doi.org/10.1002/qi.3801

Response: Thanks for your suggestion. We learned and cited the references.

---

## Author Comment (AC3)

**Responses to Reviewer #1**

**Reviewer #1**

The authors present an analysis of the seasonality of QBO effects on stratospheric water vapor in ERA5 reanalysis data, the observation-based SWOOSH data set, and CMIP models. In general, I find that this analysis provides sufficient new knowledge about the QBO impact on stratospheric water vapor to merit publication. I also think that mostly the analysis is well presented. I have some specific concerns about details of the analysis, its interpretation, and references to earlier work that I will list in the following, and which I think should be addressed before I can recommend a publication.

Response: Thank you for your comments concerning our manuscript. Those comments are all valuable and very helpful for revising and improving our paper, as well as the important guiding significance to our research. We have studied the comments carefully and made corrections which we hope meet with approval.

Introduction: I think the motivation for this study should be sharpened. From the introduction I have the impression that the main research gap is given in the following sentences: "However, it still remains unclear whether the effects of the QBO on stratospheric water vapor differ between northern winter and summer. The seasonality of the water vapor QBO signal has been seldom studied." "Seldom" would not mean never, so what is known about the seasonality and what not? And why is the difference between northern winter and summer of specific interest? I'd further appreciate that the authors formulate a hypothesis on expected impacts of the seasonality which could be based on existing knowledge on the seasonality of stratospheric water vapor and circulation and on the mean imprint of the QBO on these two quantities. Additionally I think it would be good to be more specific about why a seasonality of the QBO imprint on stratospheric water vapor would matter.

Response: Thanks very much for your positive comments. We added why it is interesting/important to investigate the seasonal cycle of the QBO on water vapor.

• "Serva et al. (2022) found that there are seasonal differences in temperature and SWV in tropical regions. In the northern summer, the temperature at 100 hPa and the WV at 85 hPa reach their peaks, while in winter, they reach their lowest levels (Serva et al., 2022). The QBO is affected by the BD circulation, and it is stronger in northern winter than in summer (Butchart, 2014). Tegtmeier et al. (2020) found that the temperature amplitude of QBO was 2 K in February of northern winter and only 0.9 K in September of summer. Similar questions naturally arise: Does the amplitude of WV QBO also undergo a similar change? What are the differences between winter and summer? The research on the seasonal differences of WV QBO not only deepens the multi-time scale understanding of the stratospheric and tropospheric coupling, but also provides a scientific basis for cross-seasonal climate prediction. This study uses more samples based on the long time series of the QBO signal in SWV and discusses the differences in SWV distribution between different QBO phases and between different seasons. Possible causes of those differences are diagnosed, and the performance of climate models in capturing the

QBO signal in WV is also evaluated (Ye et al., 2018; Ziskin et al., 2022)." (L70-81)

2a Datasets: What is the motivation for using ERA5 and SWOOSH? To which degree can these datasets be considered independent. Have observations used to build SWOOSH also been assimilated in ERA5? What may be the advantages of one or the other dataset?

Response: Thanks for your suggestion. The comparison between ERA5 reanalysis data and SWOOSH satellite observation data has been added. ERA5 reanalysis data are used because it has a longer time range.

• "In contrast to the troposphere, the WV content within the stratosphere is extremely low. Compared to the SWOOSH satellite observation data, the ERA5 reanalysis data provides a longer time span, which provides more samples for revealing the effect of QBO on SWV. There remains uncertainty regarding the performance of ERA5 reanalysis data in depicting SWV. In the ERA5 reanalysis, WV mainly assimilates in-situ humidity observations in the troposphere and satellite radiation observations that are only sensitive to humidity in the troposphere......" (L104-124)

L117: The authors write that divM "represents the eddy transport of water vapor" To my understanding it doesn't represent the full eddy transport, which is partly already included in the residual advection terms. It only appears in the case of tracers which are not inert. I have to admit my knowledge of this formalism is only partial, but other readers may also benefit from a more comprehensive discussion of these terms.

Response: You are correct. Thanks for your suggestion. This explanation has been added to the text.

• "In the case of a non-inert tracer, it doesn't represent the full eddy transport, which is partly already included in the residual term." (L156-157)

L134: "Since HALOE started from 1992, the water vapor QBO amplitude in the upper stratosphere between 1–5 hPa has increased, which is also shown in ERA5 reanalysis." To me this statement is unclear. Is the assumption that the assimilation of HALOE data is causing this increase in the datasets, or could the timing be accidental? This is related to the above comments on the independence of ERA5 and SWOOSH data. I also don't understand why the following sentence starts with "Alternately, ..." Do you mean "alternatively"? But even then, it seems that the two sentences discuss different phenomena, changes in time in the first, and changes with altitude in the second sentence.

Response: Thanks for your suggestion. This part has been modified, and the original description has been deleted to clarify.

• "In terms of data and methods, we compared ERA5 reanalysis with SWOOSH satellite monitoring data and found that ERA5 reanalysis data could reproduce the distribution pattern of SWV (Fig. 1). ERA5 reanalysis can well display the QBO signal of SWV below 10 hPa." (L164-166)

L164: The authors write that the "relationship between the QBO and water vapor [shown in Fig. 3] is to be expected" because the cold point temperature determines tropical water vapor. I'd agree that "a" relationship is to be expected, but why "this" relationship? Why would it be expected that both at 10 and 70 hPa there would be an in-phase relationship? Given the in-phase relationship at these two levels and the different vertical propagation directions of QBO winds and water vapor, would the relationship be out of-phase at levels inbetween (e.g. 20 or 30 hPa). If this is so, it would be good to mention it in order not to raise the false impression of an in-phase relationship everywhere. Please consider this issue also for point I of the summary section.

Response: Thanks for your suggestion. It has been removed in the latest revised version. We have modified the original Figure 3 to display the 30 hPa QBO index in relation to water vapor at different levels, and have calculated the lag correlation coefficients between the 30 hPa QBO index and water vapor at each level. Places related to your concerns are listed as follows.

- "Figure 4 shows the lagged correlation coefficients between the QBO index at 30 hPa and the WV at each level....." (L199-)
- "The 30 hPa QBO index exerts the greatest influence on 100 hPa water vapor at a lag of six months. During northern summer, the peak amplitude of 100 hPa water vapor under different QBO phases in tropical regions reaches ±0.12 ppm at a sixmonth lag, while in winter it reaches ±0.2 ppm." (L14-17)
- "The 30 hPa QBO index exerts the greatest influence on 100 hPa WV at a lag of six months....." (L423-)

Fig. 10: I think the results of this figure are not sufficiently discussed. It is said that "the residual circulation explains partially the water vapor variation in the tropical stratosphere" which I find to vague. What means partially? What else is important? And is it horizontal or vertical advection that matters? If possible I'd like to see a conclusion from this analysis arguing if tropical water vapor anomalies are mainly related to the upward propagation of different amounts of water vapor entering the stratosphere in different QBO phases or some other phenomenon. Similar for extratropical anomalies. This would also be the place for an attempt to explain the analysed differences between hemispheres and seasons.

Response: Thanks for your suggestion. Here we added the analysis and supplemented the meridional and vertical advection terms (Fig. S6).

• "The change of mean advection of WV is basically consistent with the tendency of WV in the tropical region. Positive and negative anomalies are observed at the lower and upper stratosphere, respectively. However, in the tropical lower stratosphere, the positive anomaly of the mean advection is smaller than that of the WV tendency (Fig.9b)." (L331-)

4b: Factors affecting the water vapor distribution: The discussion of temperature anomalies is motivated by the relevance of the cold point temperature. However, the

following paragraph discusses temperature anomalies also elsewhere. What is the motivation for this? Furthermore: The QBO influence on cold point temperature has been discussed by other studies. It would be good to provide references and discuss to what extent this study provides similar or different results.

Response: Thanks for your suggestion. Here we added motivation for this and provide references.

• "By analyzing the stratospheric temperature anomalies at different QBO phases, it can be found that only the cold temperature at the bottom of the tropical stratosphere can affect the change of SWV, while the temperature change in the middle and upper stratosphere does not directly alter the WV content." (L258-260)

L218: "as expected from thermal wind balance" It may be useful to add a reference here as not every reader may be familiar with the concept of thermal wind balance in the tropical atmosphere.

Response: Added (Allen and Sherwood, 2008). (L197, L241)

Figs. 4, 6, 7, 10: Please use consistent vertical extensions in these plots to facilitate comparison of the figures.

Response: All four figures have been modified to 150-1 hPa.

Fig. 8: The seasonal dependence of QBO-related temperature anomalies in the tropopause regions has been analysed earlier, e.g. by Tegtmeier et al. (GRL, 2020) or by Serva et al. (QJRMS, 2022). I'm almost certain there are even more papers on this, but I haven't performed a proper literature survey. Please discuss to what extent your results agree or disagree with earlier studies.

Response: We learned and cited the two references. We further analyzed the impact of the 30 hPa QBO index on the 100 hPa temperature with a six-month lag.

- "As the influence of the QBO signal gradually propagates to lower layers, the temperature anomaly with a lag of 6 months at 100 hPa is shown in Figure 7....." (L279-)
- "This is consistent with the discovery by Tegtmeier et al. (2020) that the QBO temperature amplitude is stronger in winter than in summer....." (L270-271)

L298: "This combination suggests that the QBO might be able to influence convection in this region." There have been many earlier studies on the dependence of convection on QBO phases. Please discuss to what extent your results agree or disagree with earlier studies. As the main goal of this study is to analyse the QBO-dependence of stratospheric water vapor, I'd like to see a discussion if the dependence of convection might impact the water vapour distribution. If not I'd suggest to remove this part.

Response: Thanks for your suggestion. This part has been moved to the Supplementary Materials.

Section 5: Figures 1 and 11 use different color scales. This may be useful to show the simulated signals more clearly, but it should be mentioned explicitly. Related to that

I'd find it useful to state clearly very early that for all models the signal is too weak. Potential reasons for that should be discussed. I understand the analysis presented in Fig. 14 as an attempt to identify an explanation, but I don't see a clear conclusion presented by the authors. If the tropopause temperature anomaly is crucial for the water vapour entry, wouldn't it be more straightforward to analyse how the strength of this anomaly in CMIP models relates to the simulation of the water vapour signal?

Response: As concluded earlier, variations in stratospheric water vapor are jointly influenced by the transport effect of the secondary circulation and the cold point tropopause temperature. Temperature is only one of the contributing factors. Thank you for your suggestions, we have made several revisions this time.

- "Given that the WV QBO signal in the CMIP6 models is generally weak, different color scales are used in Figures 2 and 10 to display the signal more clearly." (L352-353)
- Based on the feedback from the last reviewer, Figure 11 has been modified to depict the relationship between 30 hPa zonal wind and 70 hPa water vapor lagging by six months. "Figure 11 shows the scatter plots of QBO westerly phase minus easterly phase for the 30 hPa zonal wind index and 70 hPa WV anomalies with a lag of 6 months in deep tropics among CMIP6 models....." (L393-)

L327: "Since stratospheric water vapor has important climatic effects, evaluation of the simulated water vapor QBO by CMIP6 models is helpful in diagnosing how to improve the performance of the models (Keeble et al., 2021; Ziskin Ziv et al., 2022)." This may provide some of the motivation I was missing in the introduction. However, I find the statement very vague. What do you have in mind? Model performance with respect to what? How would it help in diagnosing how to improve it?

Response: A discussion on the simulation performance of QBO in CMIP6 models has been added to the Introduction to clarify this.

• "However, it remains a challenge to simulate the QBO in general circulation models (GCMs), with only a few GCMs being able to reproduce it. The waves need to be correctly represented to simulate a realistic QBO. Many GCMs still cannot simulate a realistic spectrum of tropical waves because of their low resolution and their deficiencies in the parameterization of small-scale gravity waves forcing (Ricciardulli and Garcia, 2000; Lott et al., 2014). Studies have suggested that an adequately fine vertical resolution (vertical grid spacing of ~500–700 m) of the troposphere and lower stratosphere is also necessary to simulate the QBO due to the forcing of some resolved waves with small vertical wavelength and the need to capture the wind shear (Richter et al., 2014b; Geller et al., 2016). In CMIP5, only five models could generate the QBO internally (Butchart et al., 2018). In CMIP6, at least 15 models now able to simulate realistic QBO-like behavior during the historical period (Richter et al., 2020)." (L49-57)

L367: Is this statement really true for CESM-WACCM-FV2? The table indicates a winter correlation of only 0.29, not higher than 0.5.

Response: Changed.

• "Only one model (AWI CM-1-1-MR) can simulate the seasonal contrast in WV distribution with the pattern correlation exceeding 0.5, although the general WV anomaly patterns show biases from the ERA5 reanalysis (Fig. S2)." (L386-388)

L423: As mentioned above, a seasonality of QBO signals in tropopause temperature has been identified in previous papers. Please indicate to what extent your findings are new

Response: Revised in several places.

- "Consistent with previous work (Tegtmeier et al., 2020)....." (L432-)
- We have also identified the impact of the 30 hPa QBO on the 100 hPa temperature with a six-month lag. "As the QBO signal propagates downward from the upper stratosphere, the 30hPa QBO index has a significant impact on the 100hPa temperature after six months, but the lagged temperature amplitude in northern summer is still smaller than that in winter." (L435-438)

L443: "This study ... finds that BD circulation change related to QBO might be a mediator bridging the QBO and water vapor." I have difficulties to understand this statement. Please be more precise. What means mediator? Why might? Hasn't it been shown clearly in this and earlier studies that QBO and stratospheric water vapour are related? So what is actually new in this finding?

Response: Thanks for your suggestion. This sentence has been modified.

"However, this study reveals the difference in SWV content regulated by QBO between northern winter and summer, and finds that QBO-related cold point temperature anomalies in the tropics affect WV distribution in the lower tropical stratosphere with a 6-month lag. QBO-related secondary circulation affects WV transport in the middle and lower tropical stratosphere, which provides a new perspective to better understand the SWV QBO signals." (L455-458)

L444: "It provides a new perspective to better understand the stratospheric water vapor QBO signals." Also this sentence is unnecessarily vague. What is this new perspective? Response: See comments above.

---

## Author Comment (AC4)

**Responses to Reviewer #3**

**Reviewer #3**

This paper presents an analysis of the effect of the QBO on stratospheric water vapor, in particular comparing boreal winter and summer. The first part of the analysis is based on ERA5 reanalysis and SWOOSH observations, the second part on CMIP6 climate model simulations. In my opinion, the main new results are: (i) differences in stratospheric water vapor between QBO westerly and easterly phases are significantly smaller in boreal summer than winter, (ii) this seasonal difference in the QBO effect is partly related to stratospheric circulation and partly to convection, (ii) current climate models have issues in simulating these seasonal differences.

Overall, I think this paper addresses an interesting question, presents valid analysis and new results, and should be suitable for publication in ACP. However, I have 2 major comments and a list of specific comments which I'd ask the authors to address before I can recommend publication.

Response: Thanks for your suggestion, which has been very helpful in improving the quality of our paper. We have carefully revised the paper based on the reviewers' comments.

**Major comments:**

**1.) Vertical propagation of QBO effect:**

This comment concerns the comparison of the QBO effect on water vapor at different levels. For instance, the paragraph starting at L156 is not clear and even somewhat misleading to me. Why is it meaningful to compare correlations between zonal wind and water vapor calculated at different levels? I see basically two pathways how the QBO can affect stratospheric water vapor: by modulating tropical tropopause temperatures and by modulating stratospheric transport (e.g. via induced secondary circulation). I'd suggest to decribe these two pathways clearly, e.g. already in the introduction.

Response: Thanks for your suggestion. We removed the comparison of water vapor and zonal wind QBO at different levels. In the revised manuscript, we described more clearly the two ways through which QBO affects SWV. That is, how QBO affects the cold point temperature of the tropopause and the transport effect of the BD circulation.

- "The dehydration effect by cold temperature in the lower stratosphere is also more effective in boreal winter than in summer. The intensity of the QBO-related secondary circulation is stronger in the boreal winter than in summer, which not only influences the cold point tropopause temperature in tropical regions but also drives the transport of stratospheric water vapor. The mean vertical transport term via the QBO-related residual circulation is the leading factor controlling the water vapor distribution in the tropical lower stratosphere." (L17-21)
- The effect of QBO on the tropopause temperature is added in Section 4.2. This effect is maximized at a lag of 6 months. The QBO-related secondary circulation distribution and its effect on water vapor transport is also diagnosed using the continuity equation.

**• Those two pathways are also summarized in the last section. (L431-444)**

The signal from modulating tropical tropopause temperatures propagates upwards and therefore comparison of regressions at different levels needs including lag times. Said that, couldn't it just be that the significant positive correlation at 70 an 10hPa just coincidentally results from the upward propagating tape recorder signal (Fig. 1) and downward propagating wind anomalies (Fig. 2), but has no deeper physical meaning? I see similar difficulties when interpreting Fig. 4, which also compares the QBO effect at different levels. Hence, I'd suggest to include such lag times for a proper comparison of the QBO effect at different levels (e.g. correlating water vapor at different levels with including lag. one OBO-index at 30hPa, see Diallo https://doi.org/10.5194/acp-22-14303-2022). If this goes beyond the scope of the paper, one could also remove Figs. 3 and 4 and the related discussion from the paper, but then clearly state early in the paper that the QBO here is always defined at 30hPa. Related to that, the respective sentences in the abstract (L15) and summary (L402) need to be clarified or removed.

Response: Thanks for your suggestion. We have modified the original Figure 3 to display the 30 hPa QBO index in relation to water vapor at different levels, and have calculated the lag correlation coefficients between the 30 hPa QBO index and water vapor at each level. See the revised Figure 4.

- "Figure 4 shows the lagged correlation coefficients between the QBO index at 30 hPa and the WV at each level......" (L199-)
- The time lag of QBO's impact is also considered this time. (L279-293).
- "The primary source of SWV is tropical tropospheric WV entering the stratosphere. Figure 4 shows that the modulation effect of 30 hPa zonal wind QBO on 100 hPa WV reaches its maximum after a lag of half a year..." (L220-)
- "As the influence of the QBO signal gradually propagates to lower layers, the temperature anomaly with a lag of 6 months at 100 hPa is shown in Figure 7....." (L279-287)
- Meanwhile, the original Figure 4,5, which depicts water vapor profiles under different QBO phases, remains scientifically meaningful. It illustrates a vertical cross-section of QBO's influence on water vapor, specifically showing the positive-negative-positive structure from bottom to top during QBO westerly phases, with notable differences between northern winter and summer. We have condensed the related content in the main text and moved the figure to the supplementary materials (Figs. S2, S3).
- "During both winter and summer, the influence of QBO on tropical stratospheric WV entry is nearly symmetrical. Under the QBO westerly phase as an example, the distribution of tropical stratospheric WV displays a sandwich structure with positive, negative and positive WV anomalies from the lower to upper layers (Fig. S2). Further, the WV anomalies in the lower stratosphere during winter are stronger than during summer (Fig. S3)." (L216-219)

The abstract and summary have been revised.

- "The 30 hPa QBO index exerts the greatest influence on 100 hPa water vapor at a lag of six months. During northern summer, the peak amplitude of 100 hPa water vapor under different QBO phases in tropical regions reaches ±0.12 ppm at a sixmonth lag, while in winter it reaches ±0.2 ppm." (L14-17)
- "The 30 hPa QBO index exerts the greatest influence on 100 hPa WV at a lag of six months, and previous studies have also discussed the lag effect of the QBO index on WV in the low stratosphere (Diallo et al., 2022; Ziskin et al., 2022)." (L423-425)

**2.) New results versus state-of-the-art:**

At some parts the paper is not very clear in what the new results and what just state-of-the-art is. For instance in the summary there are large text parts describing well-known facts regarding the QBO (e.g. L410-418), and also in the rest of the paper (see a few of my specific comments below). Sure, it is absolutely necessary to relate to previous work. But I'd recommend to say more clearly what the new findings of this paper are (my view on these is summarized in my general comment above). Perhaps a clearer structuring of the summary around these could be helpful (first stating the respective new finding, then discussing). Also, formulating related research questions in the introduction could help the reader here.

Response: Thank you for your suggestions. We have revised and refined the abstract (L10-23) and conclusions (L415-461) accordingly.

The research significance has been added to the Introduction.

• "The research on the seasonal differences of WV QBO not only deepens the multitime scale understanding of the stratospheric and tropospheric coupling, but also provides a scientific basis for cross-seasonal climate prediction." (L76-77)

**Specific comments:**

L18: I don't understand what is meant by "...dynamic transport...", probably "vertical transport"? And also the connection to the next sentence is not clear to me. Sure, in boreal winter dehydration is stronger. Please reformulate both sentences to clarify what is meant.

Response: This sentence has been relocated.

• "The dehydration effect by cold temperature in the lower stratosphere is also more effective in boreal winter than in summer." (L17-18)

L35: The "tropical path" is the "primary channel" for water vapor into the stratospheric overworld - for the lowermost stratosphere this is not clear. Please clarify.

Response: Changed.

• "The WV in the stratosphere mainly comes from the upward transport of tropospheric water vapor in the tropics....." (L32-)

L134: Overall, I find the agreement between ERA5 and SWOOSH here not too strong and would recommend to discuss the differences in more detail (e.g. the too fast upward propagation of the signal in ERA5, or the too strong dampening of the amplitude).

Response: Thanks for your suggestion. We have revised and only displayed water vapor anomalies below 10 hPa. The similarities and differences between ERA5 and SWOOSH satellite observations have been compared.

• "In terms of data and methods, we compared ERA5 reanalysis with SWOOSH satellite monitoring data and found that ERA5 reanalysis data could reproduce the distribution pattern of SWV (Fig. 1)....." (L164-)

L137: I'm wondering about the years after 2015. Why is there no clear QBO signal in water vapor during these years in ERA5? SWOOSH observations show a water vapor QBO also in these years. Pleae comment.

Response: Thanks for the feedback. We rechecked the code, and the WV QBO signal in the new plot is much clearer. (Figure 2, Figure S1)

Figure 2, caption: State the dataset used (ERA5) in the caption. Also describe the contour values and the meaning of dashed/solid.

Response: Thanks for your suggestion. Changed to "Temporal variations of zonal mean zonal wind anomalies (contours; units: m/s) and temperature anomalies (shadings; units: K) averaged over the equator ( $5^{\circ}S-5^{\circ}N$ ) with removed linear trends in the tropical stratosphere from 1960–2019 for ERA5 reanalysis. The contours are shown at  $\pm 15$  m/s and  $\pm 30$  m/s." (L191-193)

L156ff: Refer to Fig. 3 at the beginning of the text where it is discussed.

Response: Figure 3 has been revised (revised Figure 4) to show lag correlations based on the main comments, and the corresponding text has also been modified.

• "Figure 4 shows the lagged correlation coefficients between the QBO index at 30 hPa and the WV at each level......" (L199-)

L162: The weakening of the QBO signal above 5 hPa is well known, and one could refer to e.g. Baldwin et al. (2001). (This comment is related to major comment 2). Response: This section has been removed from the revised version.

L164: What relationship is to be expected? Please clarify.

Response: This section has been removed from the revised version.

Figure 4, caption: Explain the meaning of the shading in the caption.

Response: The original Figure 4 has been moved to the Supplementary Materials. Added "Dots denote statistical significance at the 95% confidence level based on Student t-test." (Fig. S2)

L258: I don't understand the intention behind this sentence "rising branch in the tropics stronger than the sinking branch in the extratropics". Tropical upwelling should always be balanced by extratropical downwelling. Please clarify.

Response: Changed to "Under the QBO easterly phase in the northern winter, the lower stratosphere vertical residual velocity anomaly shows upwelling in the tropics and

sinking in the Northern Hemisphere subtropics with the ascending branch in the tropics being stronger." (L311-313)

Figure 7: It would be good to have the same y-axis range and labels as in other figures (e.g. Fig. 4, 6) to ease comparison.

Response: The y-axis of Figure 7 is modified to be consistent with the previous figures.

Figure 7, caption: Residual circulation upwelling (!) ... And then give the units directly after "upwelling", not just at the end of the caption.

Response: Thanks for your suggestions. Revised.

• "Figure 8. (a) Residual circulation anomalies during the QBO westerly phase in the northern summer (units: 10-5 Pa/s). (b) Residual circulation anomalies during the QBO easterly phase in the northern summer (units: 10-5 Pa/s). (c, d) As in a, b but for residual circulation anomalies for the northern winter (units: 10-5 Pa/s). Dots mark the composite vertical residual velocity anomalies at the 95% confidence level. The shading is the vertical component of the residual velocities." (L323-326)

L295: There is a recent paper by Pena-Ortiz et al. (2024, https://doi.org/10.5194/acp-24-5457-2024) which demonstrates a relation between the QBO and convection in the Asian summer monsoon, of relevance for water vapor variability in the monsoon UTLS. I think it could be enlightening to relate the QBO-convection relation found here to their results.

Response: Added.

• "The QBO modulating the temperature signal on the southern side of the South Asian monsoon (i.e., the Indo-Pacific Warm Pool region) in July and August is related to the changes in equatorial clouds, which in turn affect the WV distribution in the upper troposphere and lower stratosphere of this region (Pena-Ortiz et al., 2024)." (L272-275)

Figure 9, caption: Should be "OLR" not "temperature" anomalies in L310. Response: Thanks for your suggestions. The original Figure 9 has been moved to the Supplementary Materials. Changed "temperature" to "OLR". (Fig. S5)

L312ff: The take home message from the tracer continuity analysis is not becoming very clear to me from the text here. I think the main result is that it is mainly mean advection by the residual circulation that causes the observed differences between easterly and westerly QBO phases. Please clarify the text that this is getting clearer. (This comment is related to major comment 2).

Response: Thanks for your suggestion. We have incorporated a more detailed description and provided a conclusion at the end.

• "Through the above analysis, it can be found that the SWV changes in the tropics caused by QBO are mainly produced by the mean advection term by the residual circulation, and the cold temperature changes resulting from the residual term at

the bottom of the tropical stratosphere also contributes to SWV variations." (L341-343)

L329ff: ERA5 shows a change in the QBO-related anomaly pattern around 10hPa: below there is a tape-recorder of upward propagating anomalies, whereas above there seems to be a more direct effect of transport modulated by the QBO-induced secondary circulation. This change in anomaly pattern is only visible in a few models (e.g. CESM2-WACCM). I find this an important difference between ERA5 and CMIP6 models and would recommend to discuss it.

Response: Thanks for your suggestion. We added this discussion.

• "ERA5 and SWOOSH show a change in the QBO-related anomaly pattern around 10 hPa. Below 10 hPa there is a tape-recorder of upward propagating anomalies, whereas above 10 hPa there seems to be a more direct effect of transport modulated by the QBO-induced secondary circulation (Fig. S1)." (L360-363)

L351: But even for this model the agreement with ERA5 is not very high. Response: Changed.

• "The WV correlation coefficient between models (CESM2-WACCM-FV2 and MRI-ESM2-0) and the ERA5 reanalysis exceeds 0.8 (Table 1), but even for those two models the biases relative to ERA5 is not little." (L375-377)

L367: Table 1 says that CESM2-WACCM-FV2 for boreal winter has a correlation coefficient of 0.29. So I'm wondering why it is mentioned here as one of the two models where correlation is higher than 0.5.

Response: Changed to "Only one model (AWI CM-1-1-MR) can simulate the seasonal contrast in WV distribution with the pattern correlation exceeding 0.5..." (L386-388)

L375ff: For the analysis here, a QBO index at 50hPa is used. Before (L102ff) it was argued that in this paper a 30 hPa-index is used as only at such high level all models have a sufficiently significant QBO variability. This seems somewhat contradictory to me. Please clarify.

Response: Thanks for your suggestion. Referring to the updated Figure 4, Figure 11 here revised to show the relationship between the 30 hPa QBO index and the 70 hPa water vapor lagging by six months.

• "Figure 11 shows the scatter plots of QBO westerly phase minus easterly phase for the 30 hPa zonal wind index and 70 hPa WV anomalies with a lag of 6 months in deep tropics among CMIP6 models......" (L393-)

L402: The enhanced correlation for lower and upper stratosphere seems method-related to me (see my major comment above). I'd remove this statement here or clarify. Response: Thank you for your suggestion.

• "The 30 hPa QBO index exerts the greatest influence on 100 hPa WV at a lag of six months, and previous studies have also discussed the lag effect of the QBO

index on WV in the low stratosphere (Diallo et al., 2022; Ziskin et al., 2022)." (L423-425)

L410ff: In my opinion, most of this paragraph is well-known facts regarding the QBO and induced secondary circulation. Only the last sentence (starting L418) describes the new results of the present paper. I'd recommend to restructure and shorten, so that the focus is on new results. (This comment is related to major comment 2).

Response: Thank you for your suggestion. The revised version focuses on describing the new findings of this paper.

• "The difference in the secondary circulation associated with QBO between winter and summer is compared....." (L439-)

L420: ... controlling factor ... in the tropical lower stratosphere...

Response: Changed to "...in the tropical lower stratosphere." (L444)

Technical corrections:

L13: seasonal differenceS

Response: Changed to "differences". (L12)

L58: The reference is sometimes written "Ziskin Ziv et al. (2022)", sometimes "Ziskin et al. (2022)" (e.g. L60). Please use the same citation label.

Response: Changed to "Ziskin et al. (2022)". (L64)

L192: ... anomalies ... are ... Response: Changed. (L219)

L212: regulates

Response: Changed. (L236)

L206: ... at 100hPa during winter ... (would be good to add the season here).

Response: It has been removed in the latest revised version.

L247: anomalously strong upwelling ... anomalously strong downwelling ...

Response: Removed.

L256: in the Northern hemisphere suptropics

Response: Changed. (L312)

L427: I'm not entirely sure what is meant here. Do you mean: "The influence of OLR on the tropopause cold point temperature in summer is opposite to the tropical stratospheric temperature anomalies related to the QBO secondary circulation, which ..."

Response: It has been removed in the latest revised version.